# Compounded effects of long-term warming and the exceptional 2023 marine heatwave on North Atlantic coccolithophore bloom dynamics

Thibault Guinaldo[1] and Griet Neukermans[2,3]

[1]Centre National de Recherches Météorologiques, Université de Toulouse, Météo-France, CNRS, Toulouse, France
[2]Ghent University, Biology Department, MarSens Research Group, Krijgslaan 281 – S8, 9000 Ghent, Belgium
[3]Flanders Marine Institute, InnovOcean Campus, 8400 Ostend, Belgium

**Correspondence:** Thibault Guinaldo (thibault.guinaldo@meteo.fr)

**Abstract.** The North Atlantic is undergoing rapid ecological evolution under the influence of both long-term warming and the increasing frequency of extreme marine heatwaves. In 2023, the North Atlantic experienced record-breaking sea surface temperature anomalies, exceeding +5°C regionally and lasting several months. Using 25 years of satellite-derived particulate inorganic carbon data (1998–2023), we assess the response of coccolithophores blooms across two biogeographical boundaries: the Celtic Sea and the Barents Sea. We show that the 2023 MHW led to reduced bloom intensity and fragmentation in the Celtic Sea, while leading to record-high intensity and extent in the Barents Sea. These contrasting responses are modulated by long-term SST trends, upper-ocean stratification, and polar front shifts. Our findings suggest a spatial shift of coccolithophore blooms with potential implications for the carbon cycle under long-term warming and stratification.

## 1 Introduction

During boreal spring and summer, large parts of the North Atlantic Ocean are transformed into shades of color, indicating the occurrence of phytoplankton blooms. Among these, coccolithophores, particularly *Gephyrocapsa huxleyi* (Bendif et al., 2023) the most abundant species, form extensive summer blooms, which may weaken the ocean sink for atmospheric carbon dioxide (Shutler et al., 2010; Kondrik et al., 2018). During the decline phase of these blooms, the overproduction and detachment of calcite plates (coccoliths) color the surface waters a distinctive milky-turquoise, detectable by optical satellite sensors (Ackleson et al., 1988; Moore et al., 2012; Neukermans and Fournier, 2018). As photosynthetic organisms, coccolithophores contribute 1-10% to global ocean primary production (Poulton et al., 2007) and about 50% to the deep ocean flux of particulate inorganic carbon (PIC; Neukermans et al., 2023). Coccolithophores thus contribute to both the organic carbon pump and the carbonate counter pump mechanisms (Neukermans et al., 2023). Finally, coccolithophores are a major producer of dimethylsulfide (Malin et al., 1993), that can promote the formation of marine clouds with important implications for climate regulation

(Fiddes et al., 2018; Mahmood et al., 2019).

Optical satellite observations, available since the late 1970's, reveal a poleward expansion of *G.huxleyi* blooms (Winter et al., 2014), at a particularly rapid rate in the Barents Sea (Neukermans et al., 2018). This shift of *G.huxleyi* blooms may be driven by increased advection of water masses in which *G.huxleyi* is already established (Oziel et al., 2020), and/or by improved bloom-ing conditions toward higher latitudes, including increasing water temperatures (Winter et al., 2014; Beaugrand et al., 2013; Rivero-Calle et al., 2015; Neukermans et al., 2018), or increasing water column stratification giving competitive advantages for *G.huxleyi* (Neukermans et al., 2018).

Over the past 40 years, oceans have absorbed approximately 91% of anthropogenic excess heat (Von Schuckmann et al., 2020), leading to significant increases in ocean heat content and raising concerns about an accelerated warming (Li et al., 2023; Minière et al., 2023). Globally, SSTs have risen by an average of 1.0°C between 1850-1900 and 2015-2024 (Forster et al., 2025). This long-term warming trend, combined with internal variability, results in anomalously high SSTs known as marine heatwaves (MHW, Hobday et al., 2016; Oliver et al., 2021). These events have become more frequent and intense, with the North Atlantic emerging as a hotspot, particularly at high latitudes (Oliver et al., 2018). These extremes can extend verti-cally and increase environmental pressure on marine ecosystems altering trophic functions, thus leading to economic impacts (Smith et al., 2021, 2023). These effects are exacerbated by a combination of biogeochemical or atmospheric conditions known as compound events (Zscheischler et al., 2018; Burger et al., 2022; Le Grix et al., 2022) causing an irreversible state for marine communities (Santana-Falcón et al., 2023; Wernberg et al., 2025).

The Atlantic Ocean has experienced some of the most pronounced ocean heat content increases (Cheng et al., 2022) that has led to unprecedented marine heat extremes, particularly affecting the Northwest European Shelf (Fig.1a; Guinaldo et al., 2023; Simon et al., 2023). In 2023, a record-breaking marine heatwave developed, resulting in SST anomalies exceeding +5°C across broad areas of the shelf for 16 days in June (Berthou et al., 2024). In fact, the entire North Atlantic has reached record-level SSTs explained by anomalies in the air-sea heat fluxes, amplified by anthropogenically driven stratification of the upper ocean and shoaling of the mixed layer depth (MLD; Guinaldo et al., 2025; England et al., 2025).

Several studies have assessed the causes of MHW and their impacts on phytoplankton communities based on chlorophyll-a measurements (Sen Gupta et al., 2020; Capotondi et al., 2024) and some have documented impacts of MHWs on phytoplankton blooms using remotely sensed chlorophyll-a observations (Cheung and Frölicher, 2020; Arteaga and Rousseaux, 2023; Cyr et al., 2024). In this study, we investigate how the extreme 2023 MHW event impacted coccolithophore bloom dynamics across the North Atlantic Ocean, using remotely sensed PIC observations. We focus on two biogeographical limits, the Celtic Sea and the Barents Sea, respectively representing the trailing (or equatorward) edge and the leading (or poleward) edge of *G.huxleyi* bloom distribution in the North Atlantic Ocean (Winter et al., 2014). Using 25 years of ocean colour satellite data (1998–2023), we assess changes in the phenology of *G.huxleyi* blooms (including timing and intensity), as well as spatial extent, and contextualise them within long-term trends.

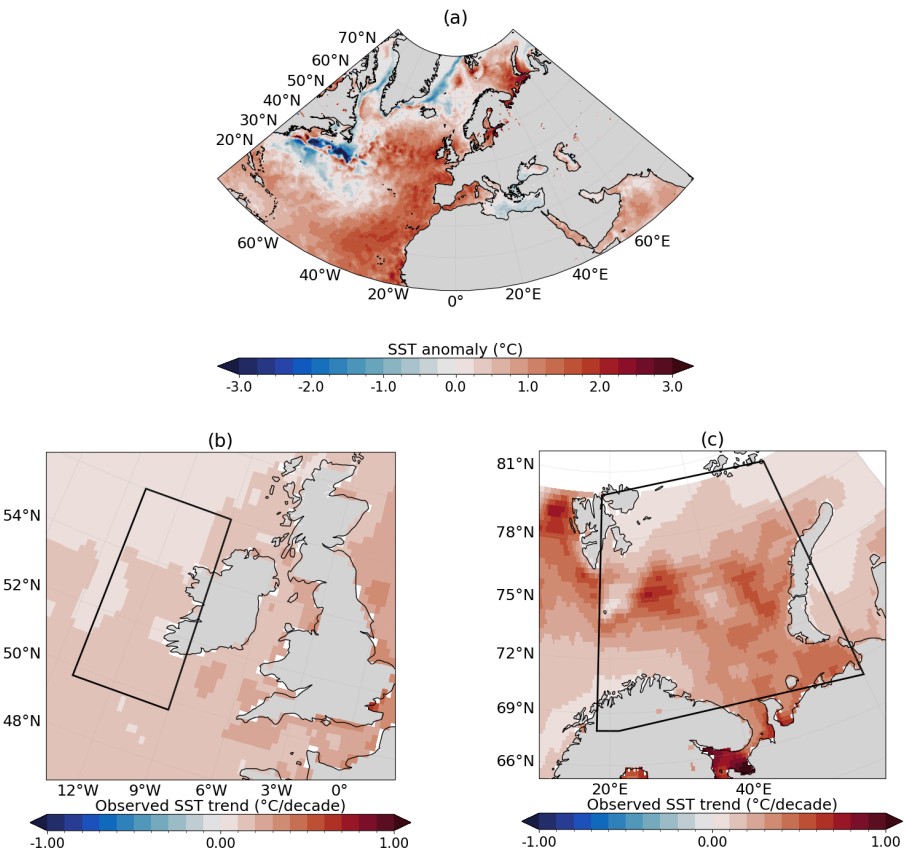

**Figure 1. Maps of SST anomalies for May-June 2023 in the North Atlantic and SSTs trends over the study sites.** (a) May-June averaged SSTs anomalies compared to the corresponding 1991-2020 period. Observed SST linear trend expressed in Celsius degrees by decade, computed from OSTIA over the period 1958-2023 for (b) the Celtic Sea and (c) the Barents Sea. Black boxes indicated the study sites chosen.

## 2 Results & Discussions

### 2.1 Environmental conditions

To evaluate the impact of MHW on *G.huxleyi* blooms, we examine impacts on the three most influential environmental variables that characterize the ecological niches of coccolithophore species, namely SST, Photosynthetically Active Radiation (PAR), and the depth of the mixed layer (MLD), an indicator for the water column stratification (see Sect.A1.4; O'Brien et al., 2016). For *G.huxleyi*, the optimal SST range was found to be situated between 6 and 16°C, optimal PAR between 35 and 42 Einstein.m$^{-2}$.day$^{-1}$, and optimal MLD between 20 and 30 m (O'Brien, 2015). These ranges were extracted from the realized ecological niche of *G.huxleyi* (i.e. the environmental conditions under which it can be observed) set up by O'Brien (2015), based on a global compilation of in situ measurements of coccolithophore species abundance and diversity (O'Brien et al.,

2013).

In 2023, both the CS and BS experienced exceptional MHWs beginning in spring (Fig.2a-b). Annual mean SST anomalies reached +0.67°C in the CS, peaking in June, and +0.92°C in the BS, peaking in August. Maximum daily SST anomalies reached +3.9°C (corresponding to 17°C) in CS and +3.3°C (corresponding to 8.8°C) in BS with warmer SSTs locally.

These MHWs were exceptional in both intensity and duration, lasting 82 days in the CS and 120 days in the BS. This occurred in a context of unprecedented global ocean heat anomalies (Terhaar et al., 2025) with a particular warming signature over the North Atlantic ocean (England et al., 2025; Guinaldo et al., 2025). These events were boosted by the long-term warming trend, particularly in the BS, where the warming trend is more than twice the global average (Fig.1c).

Likewise, PAR in CS was strong in May-June with values surpassing 42 Einstein.m$^{-2}$.day$^{-1}$ (upper-range of the optimal conditions for *G.huxleyi* with thresholds established from the study of the species' realized ecological niche; see Sect.A1.4 and O'Brien (2015)) with conditions becoming more favorable in July later (Fig.2c). These variations are primarily influenced by the atmospheric conditions, specifically cloud cover. In June, a persistent high-pressure system over Fennoscandia (Fig.A1) led to exceptionally weak wind conditions (Fig.A2) and low cloud cover (Fig.A3) but increased toward climatological values later. In BS, PAR was exceptionally high compared to the summer climatology allowing sufficient sunlight to reach the surface ocean for photosynthesis throughout summer (Fig.2d). These results are influenced by the cloud cover over BS where a large portion of the sea experienced significant clear-sky conditions during summer (Fig.A3).

In 2023, MLD dynamics followed the usual seasonal cycle in both basins, with winter deepening and summer shoaling, modulated by atmospheric conditions (de Boyer Montégut et al., 2004, Fig.2a-b). In winter, the North Atlantic Oscillation (NAO) influences vertical turbulent mixing through the atmospheric conditions associated with its positive phase, which typically include enhanced westerlies and increased storm activity over the North Atlantic (Hurrell et al., 2003). Even at the northern edge of the North Atlantic, the BS atmospheric and oceanic internal variability responds to both positive and negative NAO conditions (Levitus et al., 2009; Chafik et al., 2015). In contrast, summer conditions favor the likelihood of high-pressure blocking systems over northern Europe (Rantanen et al., 2022; Rousi et al., 2022), characterized by weak winds and high solar radiation (Fig.A1 & Fig.A2 & Fig.A3). These favour upper-ocean warming, weak winds and shallow MLD, leading to MHW development (Holbrook et al., 2020). This relationship is particularly visible in CS during June where the upper-ocean remained stratified in response to persistent high-pressure systems (Fig.A1).

In the CS, SSTs remained around 13°C from May to mid-June, in the optimal thermal range for *G.huxleyi* blooms (6°C-16°C, see Sect.A1.4 & Fig.2a). However, from June onwards SSTs frequently exceeded 16°C, with localized peaks over 20°C at the peak of the MHW (Berthou et al., 2024). This period was followed by a temporary deepening of the MLD greater than 40m due to the return of westerly winds in July and August (Fig. A2). A second MHW developed in September associated with the return of both SST and MLD (< 40m) to favorable bloom conditions.

Conversely, the BS exhibited less temporal variability throughout summer and maintained temperature within the 6°C-16°C range (summer mean: 6.8°C) and exhibited persistent shallow MLD (summer mean : 18 m) consistent with weaker than normal

or close to normal winds (Fig.A2), providing sustained conditions favorable for bloom development from July to early October (Fig. 2b, see Sect.A1.4).

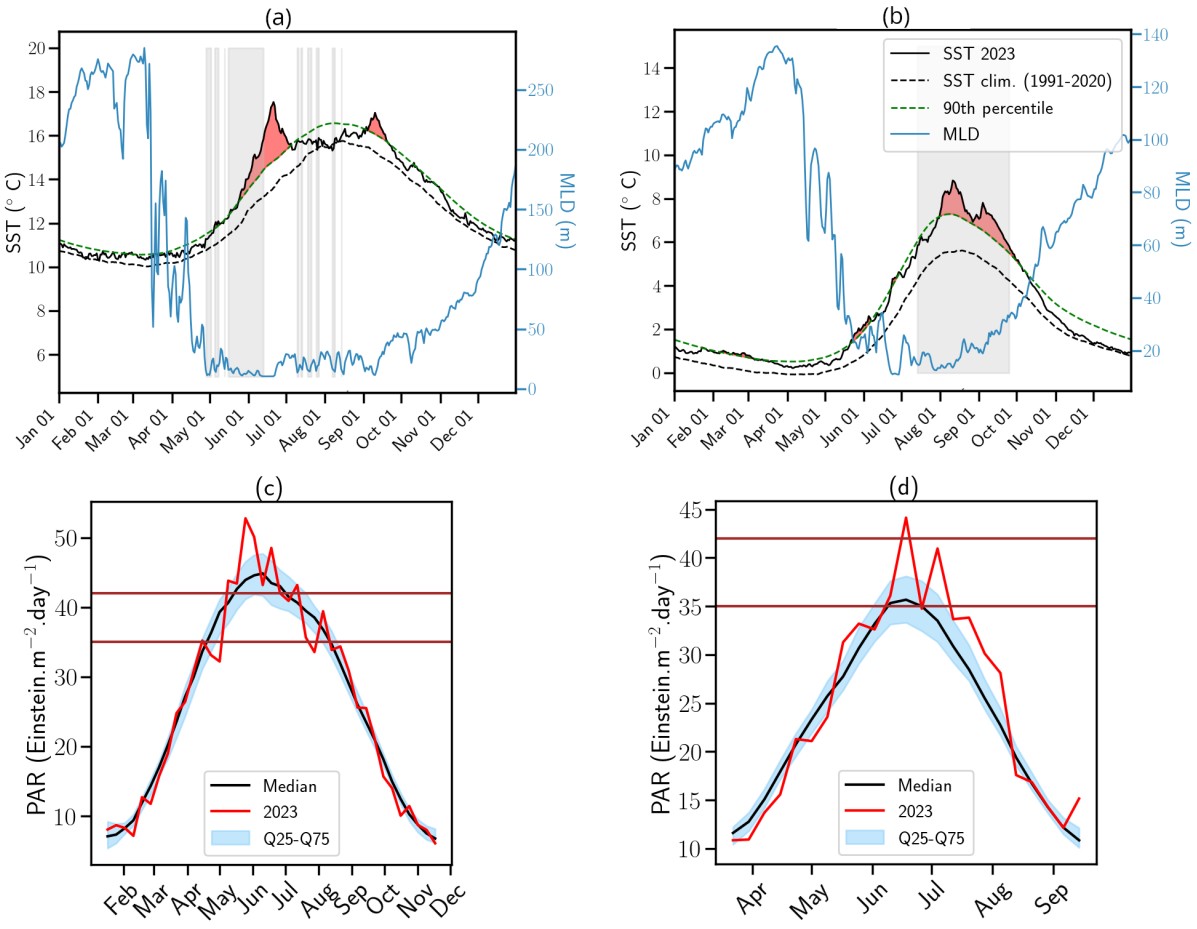

**Figure 2. Daily spatially averaged SST, MLD and PAR variables for 2023.** Spatially averaged SST (black solid line) and MLD (blue solid line) anomalies for 2023 in (a) the Celtic Sea and (b) the Barents Sea. The black dashed line represents the climatological SST averaged over each basin for the period 1991–2020, while the green dashed line marks the 90th percentile threshold for MHWs, as defined by (Hobday et al., 2016). Red shading indicates periods of MHWs, while grey shading highlights conditions favorable to *G.huxleyi* blooms based on optimal ranges for SST and MLD in the species' realized ecological niche (see Sect.A1.4; O'Brien, 2015). Spatially averaged PAR for 2023 in (c) the Celtic Sea and (d) the Barents Sea. The vertical brown lines inform on the optimal range for *G.huxleyi* blooms (see Sect.A1.4).

## 2.2 Bloom dynamics and characteristics

Satellite-derived PIC time series revealed contrasting bloom variations reflecting the SST and MLD conditions in both basins. In the CS, the bloom followed a typical seasonal evolution, intensifying from April, peaking in June at 0.30 mmol.m$^{-3}$ (below the climatological median of 0.43 mmol.m$^{-3}$; Fig.3a), and ending early July. An unusual secondary bloom emerged in

August-September, coinciding with positive SST anomalies and reached 0.38 mmol.m$^{-3}$, above the climatological Q75 (0.24 mmol.m$^{-3}$, Fig.3a). While late-summer and autumn blooms have been observed historically (September maximum over the 1997-2022 period: 0.61 mmol.m$^{-3}$, Fig.3a), the 2023 event exceeded the interquartile climatological range. The 2023 mean bloom surface extent in the CS reached a record 46 460 km$^2$ (Fig.3c), representing a 30% increase over the previous record (2007) and an 85% increase relative to the 1998-2021 mean. However, the maximum extent (126 163 km$^2$) remained close to the 1998–2010 mean (126 345 km$^2$) suggesting persistent but unevenly distributed blooms across the CS (Fig.A4). These levels were anomalously high in 2022 and 2023 (mean surface extent anomaly: 25 551 km$^2$; maximum surface extent anomaly: 25 864 km$^2$) and showed a significant correlation with spring–summer SSTs. Correlation coefficients reached 0.76 and 0.70 for mean and maximum surface extent, respectively, in relation to May–June SSTs, and 0.98 and 0.77 for mean and maximum surface extent, respectively, in relation to July–August–September SSTs (all p_value < 0.01). These strong correlations are consistent with the multi-year trend of increasing spring–summer SSTs (Fig. 3e; Fig. A5a). The significant correlation between summer SST and bloom extent suggests an important role for sustained surface warming in modulating bloom dynamics in this region.

In the BS, an exceptional summer bloom in 2023 peaked at 6.25 mmol.m$^{-3}$ (much higher than the climatological median value of 1.15 mmol.m$^{-3}$, Fig.3b), reaching values similar to the previous record set in 2022 (6.16 mmol.m$^{-3}$). The 2023 bloom occurred simultaneously with the development of the MHW in June, providing SST and MLD conditions ideal for blooms (see Sect.A1.4 and Sect.2.1). The bloom extent reached record highs of 833 561 km$^2$ in 2023; much higher than the mean value of 164 188 km$^2$ and 18% higher than the previous record of 703 174 km$^2$ set in 2022 (Fig.3d). This reflects a northeastward expansion of coccolithophores in the BS linked to the shifting Polar Front (Fig.A6, Fig.A7, Neukermans et al., 2018; Oziel et al., 2020), with blooms covering 59% of the basin in 2023, a new record for expansion. Over the past 25 years, Locally Estimated Scatterplot Smoothing regression (LOESS, Cleveland, 1979) reveals significant positive trends in bloom extent: 3 137 km$^2$.year$^{-1}$ for the mean and 12 346 km$^2$.year$^{-1}$ of the maximum spatial (p_value<0.01) with a record maximum area exceeding by 392 000 km$^2$ the climatological value (Fig.3f).

In the BS, the increase in bloom extent was also strongly and significantly correlated with summer SSTs (Fig. A5b), with correlation coefficients of 0.95 and 0.96 for mean and maximum surface extent, respectively, in relation to July–August–September SSTs (all p_value < 0.01). This highlights the role of warming in driving these changes. Two distinct processes contribute to this warming: long-term ocean temperature increase, especially pronounced at high latitudes, and the enhanced influence/inflow of Atlantic Water, commonly referred to as "Atlantification. (Årthun et al., 2012; Polyakov et al., 2017). In the BS, Atlantification encompasses not only a northward shift of the Polar Front, but also the progressive warming, increase of salinity, loss of winter sea ice, and modification of stratification conditions of waters. While these changes are strongest south of the front, modified Atlantic water increasingly reaches the northern, traditionally Arctic domain, particularly during ice-free winters (Årthun et al., 2012). To disentangle these contributions, we tracked the annual position of the Polar Front, a proxy for the influence of Atlantic water (Fig.A7a, Neukermans et al., 2018). While the mean position of the front has shifted approximately 95 km northward since the early 2000s (Fig.A7b), its position has stabilized over the past two years following a strong southward shift from 2016 to 2020, with extension occurring primarily toward the east rather than the north (Fig.A6 & A7a). In parallel, ocean

warming continued, with a long-term trend of +0.26°C per decade (Fig.1c & A5b). The 2023 MHW further amplified this trend, producing localized SST anomalies across the BS basin and leading to one of the warmest boreal summer anomalies in the Barents Sea (Fig.A5b). A significant correlation between bloom surface extent and SSTs confirms the influence of gradual warming and interannual variability on coccolithophore proliferation. However, the polar front is only a proxy of the process of atlantification and weak correlation between the position of the thermal front (western basin : 0.45, p_value <0.05; eastern basin: 0.35, p_value = 0.07) and the leading edge of coccolithophore bloom distribution (Fig.A8) suggests an important role for SST which should exceed the 6°C limit for *G.huxleyi* to proliferate.

### 2.3  *G.huxleyi* bloom trends in the satellite era (1998-2023)

Basin-averaged analysis can disregard spatial features such as the repartition and the evolution of blooms across each basin. Here, we focus on the spatial features of the blooms.

In the CS, no significant trend in summer PIC maxima was detected (Fig.A9a). In contrast, the BS exhibited a northeastward shift in summer maximum concentrations (Fig.A6 & A8). While the western BS shows limited front variability and no consistent trend, the eastern BS is characterized by high interannual variability and a long-term northward shift of 300 km for the northernmost position of the bloom and an eastward shift of 155 km for the latitudinal mean position of the bloom. Even though the latitudinal mean front position has regressed since 2016, a value close to the maximum reached in 2023 (Fig.A8), exhibiting a spike in the northward maximal expansion in 2022 and 2023 (Fig.A6). This spatial reorganization of plankton distribution in the Barents Sea has been associated with 'Atlantification', which in turn enhances blooms of temperate phytoplankton such as *G.huxleyi* through bio-advection (Oziel et al., 2017). However, this phenomenon does not fully explain the exceptional bloom observed in 2023 even if the interannual variability in the position of the polar front is accompanied by shifts in PIC maxima (e.g. 2004, 2010; Fig.A6 & A7 & A9b).

Phenological analysis reveals contrasting bloom dynamics in the two regions. In the CS, bloom timing, duration and intensity exhibited marked interannual variability (Fig.4a). On average, blooms last 90 days, ranging from 21 to 200 days with onset typically in mid-April and decline by mid-July. For the last three years (2021-2023), bloom duration increased by 130% relative to the 1998-2020 mean (Fig.4a), reflecting a shift in seasonal conditions due to recurrent late-summer warming (Fig.A5a). However, average peak PIC concentration declined by 20% during the last three years compared to the 1998-2020 period (2021-2023 average compared to the 1998-2020: 1.4 mmol.m$^{-3}$). This recent decline gives a negative trend in bloom intensity over the 25-year period (-0.03 mmol.m$^{-3}$ per year), although this trend is not statistically significant (p_value= 0.07; Fig.A9a). In contrast, the BS exhibited more consistent bloom timing, with mean bloom duration of 70 days [35–118], beginning in mid-June and ending in mid-August (Fig.4b), reflecting a shorter seasonal window for bloom development. No temporal trend in bloom timing was detected between 1998 and 2023. However, bloom intensity increased significantly in recent years. A new record daily PIC value of 16.5 mmol.m$^{-3}$ was observed in 2023, nearly double the previous record set in 2022 (8.6 mmol.m$^{-3}$). The mean peak intensity over 2021-2023 reached 12.5 mmol.m$^{-3}$, compared to the 3.2 mmol.m$^{-3}$ over 1998-2020. This surge resulted in a significant positive trend in bloom intensity (0.17 mmol.m$^{-3}$ per year, p < 0.05, Fig.A9a).

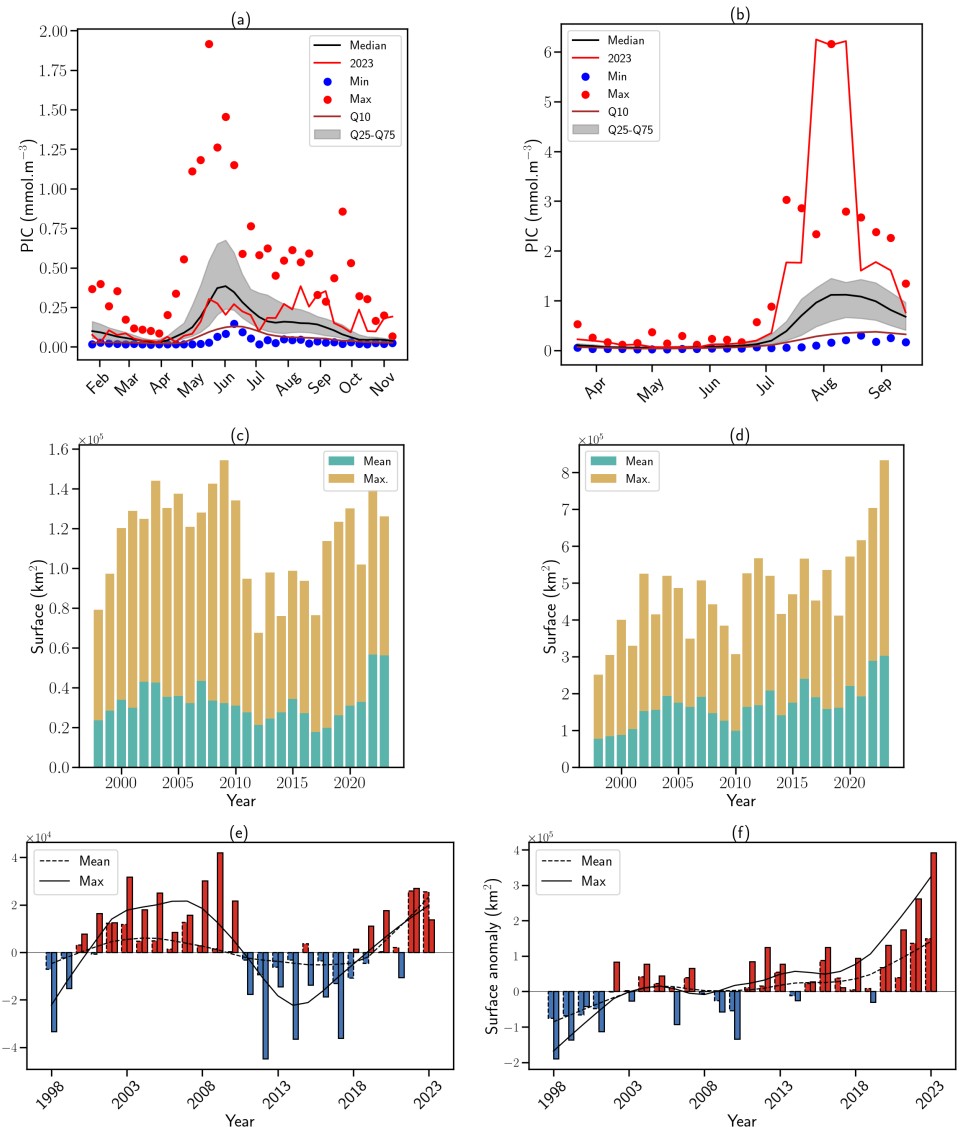

**Figure 3. *G.huxleyi* bloom phenology and surface extent in the Celtic and Barents Seas over the satellite record (1998-2023).** Seasonality in satellite-derived PIC concentration in 2023 (red line) compared to the 1998-2020 climatology (black line) in the (a) Celtic Sea and (b) the Barents Sea. Blue (resp. red) dots indicate minimum (resp. maximum) PIC concentration in the 1997-2022 climatology. Grey shading represents the 25-75 interquantile range. Brown lines represent the 10th percentile. Maximal (yellow bars) and mean (green bars) bloom spatial extent in the (c) CS and (d) BS. Corresponding surface extent anomalies (seasonal mean are in solid contour and seasonal maximum in dashed contour) for (e) CS and (f) BS. Anomalies are computed relatively to the 1998-2018 climatological period. Red bars indicate positive anomalies, while blue bars indicate negative anomalies. The lines indicate the corresponding 10-year LOESS trend.

Bloom development also depends on upper-ocean stratification, which relates to nutrient and light availability, as well as mixing. Both the CS and BS exhibit long-term trends toward stronger stratification (Fig.A10 & A11). In the CS, this trend is driven by temperature (Fig.A10), while in the BS, the trend is mostly driven by changes in salinity, with temperature playing a secondary role (Fig.A11). Positive stratification anomalies were recorded in both regions in 2023, with the CS reaching record levels, which supported favourable conditions for *G.huxleyi* (see Sect.A1.4).

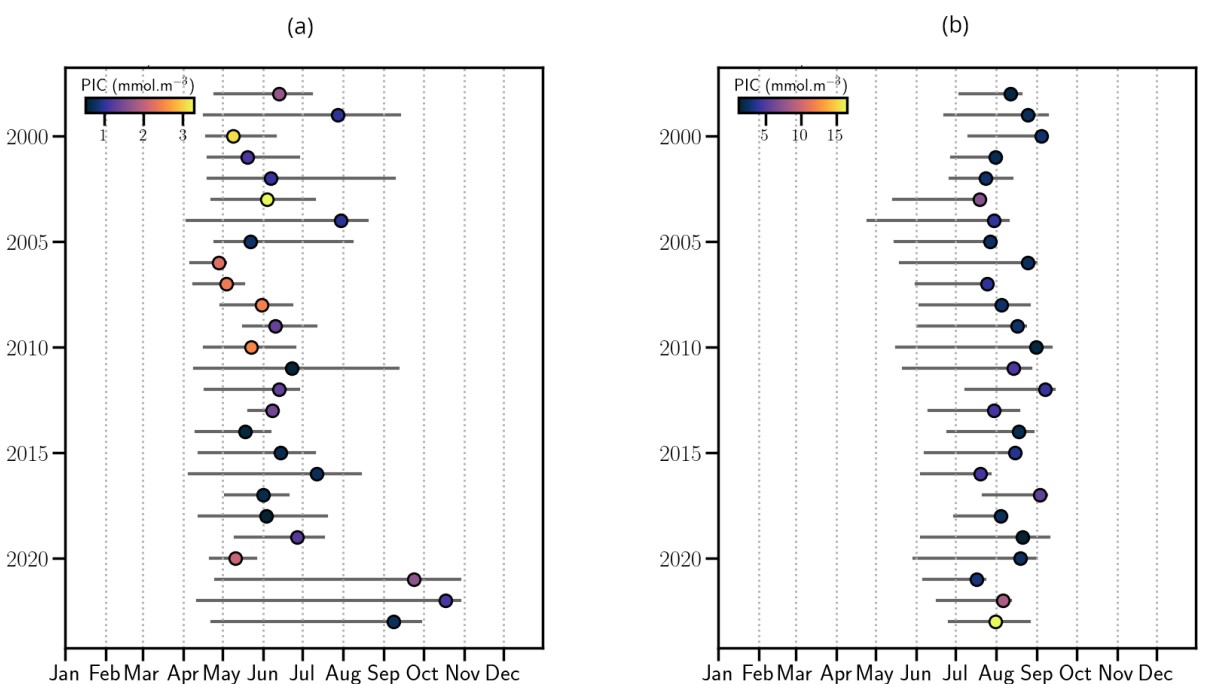

**Figure 4. Phenology of coccolithophore blooms in the North-eastern Atlantic ocean (1998–2023).** Start and end dates of blooms with maximum PIC, colored by peak PIC concentration (mmol.m$^{-3}$) in the (a) CS and (b) the BS.

## 3 Conclusions

The year 2023 was marked by extreme surface ocean temperatures extending across the North Atlantic over a prolonged period. In particular, the Northeast Atlantic experienced extremes that were both anomalously intense and prolonged (Berthou et al., 2024). Using ocean-colour satellite sensors, our results reveal how calcifying phytoplankton respond differently at biogeographic limits: degradation at the trailing edge (Celtic Sea) and amplification at the leading edge (Barents Sea).

In the Celtic Sea, recent anomalies reflected prolonged bloom duration but reduced intensity. The primary bloom was attenuated when both SSTs and PAR exceeded the optimal thermal limit of *G.huxleyi* (>16°C), and although a secondary bloom

developed in late summer, overall bloom intensity declined over the 1998-2023 period. A trend of prolonged but weaker blooms was observed in 2021–2023, combined with increased stratification which may indicate a shift toward less favorable conditions for *G.huxleyi* to bloom.

In contrast, the Barents Sea exhibited an unprecedented bloom in 2023, both in intensity and spatial coverage while the bloom timing remained stable. The persistence of favourable SSTs and PAR, coupled with a stable water column and shallow MLDs, sustained intense bloom conditions over three months. This intensification is consistent with long-term Atlantification trends (Oziel et al., 2020; Neukermans et al., 2018), a northward migration of the polar front (Fig.A7), and haline-driven stratification (Fig.A11). The Barents bloom peaked at 16.5 mmol·m$^{-3}$, more than five times the 1998–2020 mean, confirming the development of blooms in high-latitude regions (Hutchins and Tagliabue, 2024) and the poleward shift of temperate phytoplankton in the Barents Sea (Neukermans et al., 2018).

These contrasting phenological responses illustrate the sensitivity of coccolithophore bloom dynamics to both short-term (MHWs) and long-term (warming, stratification) regional environmental changes. The observed phenology aligns with previous phenological studies (Hopkins et al., 2015) but suggests a change since 2021, with poleward shifts in bloom intensity and extent potentially indicative of a regime transition. However, the weak correlation between the polar front and leading-edge bloom extent and the strong correlation between SST anomaly and bloom surface extent (see Sect.2.2) suggest that while Atlantification supports expansion, MHWs are also important in reaching thermal thresholds for bloom development. These changes need further confirmation, particularly with additional data from diverse regions undergoing similar MHWs (e.g. North Sea). Whether these changes represent a persistent regime shift remains uncertain at this point and will require analysis of a longer time series.

Nevertheless, several limitations must be acknowledged. First, our estimates of bloom spatial extent may be underestimated due to the spatial averaging inherent in satellite observations and due to masking by cloud cover at high latitudes. Future studies could explore the use of more precise spatial masking techniques to address this or rely on potentially improved retrieval of PIC from hyperspectral ocean-colour sensors such as the recently launched Plankton, Aerosols, Clouds and Ecosystems (PACE) mission (Werdell et al., 2019).

Our analysis was limited to surface ocean PIC concentration, detectable from ocean colour satellites. The lack of vertically resolved observational data constrains our ability to capture the vertical distribution and PIC standing stock of these blooms. This limitation may be overcome by applying statistical relationships extrapolating surface observations to sub-surface (e.g. Balch et al., 2018).

Coccolithophores are microscopic, calcifying phytoplankton that contribute substantially to marine primary production and global carbon cycling through both the organic carbon and carbonate pumps (Rost and Riebesell, 2004; Neukermans et al., 2023). Ocean acidification, driven by increased $CO_2$ uptake, reduces carbonate ion availability and lowers pH, creating challenging chemical conditions for calcifying organisms (Riebesell et al., 2000; Iglesias-Rodriguez et al., 2008; Terhaar et al., 2020). Laboratory experiments show that impacts are dependent on the species. *G. huxleyi* exhibits decreased calcification and lighter coccoliths under elevated $CO_2$, while other species may be more resilient but remain vulnerable to future acidifi-

cation (Meyer and Riebesell, 2015). Responses also depend also on $CO_2$ enrichment which may allow partial compensation of calcification (Fukuda et al., 2014). Global models project regionally heterogeneous effects, with some areas experiencing enhanced calcification due to carbon limitation alleviation, but a general decline is expected above 600 $\mu$atm $CO_2$, with studies on long-term impacts suggesting progressive damages (Krumhardt et al., 2019; Tong et al., 2018). Considering ocean acidification alongside warming and Atlantification provides essential context for interpreting the observed long-term declines in North Atlantic coccolithophore blooms.

Additionally, the evolution of water column stratification plays a key role in promoting blooms with a clear signal in the North Atlantic. These dynamics, including the vertical variation of the summertime mixed-layer depth (Sallée et al., 2021), may reduce both light and nutrient availability, and also have implications for carbon export, a critical function of calcifying species. Coccolithophore blooms can influence the regional ocean carbon cycling by modifying surface $pCO_2$ through the combined effect of primary production and calcification air–sea $CO_2$ exchange, and carbon export and deep particle fluxes through the calcite ballast effect (Shutler et al., 2013; Delille et al., 2005; Rigual Hernández et al., 2020; Klaas and Archer, 2002). Understanding and disentangling these influences on carbon cycling now and in the future is therefore crucial, especially as any potential long-term weakening of the ocean carbon sink may compound with short-term decline associated with MHW events. (Müller et al., 2025).

The changes observed in 2023 are an extreme signature of multi-annual variability superimposed on long-term trends. There is a need to disentangle the contributions of internal climate system variability, such as decadal variability, from the impacts of anthropogenic climate change. This will increase our capacity to assess extreme but plausible events such as the record SSTs in 2023-2024 (Terhaar et al., 2025) and anticipate their consequences. Although attribution science has made substantial progress in recent years (Stott et al., 2016; Ribes et al., 2020; Faranda et al., 2024), these developments have focused primarily on terrestrial and atmospheric variables. In ocean biogeochemistry, formal attribution frameworks are still lacking, mainly because they require multi-decadal observations and dedicated model ensembles, which are not yet available for PIC variability. Developing attribution capabilities for marine biogeochemical systems would therefore require both multi-scale observation networks providing robust initial conditions and/or a solid observational "baseline" and improved modelling frameworks able to resolve subsurface dynamics and multi-stressor interactions (Gregg and Casey, 2007; Nissen et al., 2018; Krumhardt et al., 2019). Establishing such tools and datasets would be essential before the respective roles of internal variability, extreme events, and long-term anthropogenic forcing can be formally disentangled.

*Data availability.* OSTIA SST data are publicly available for download from the UK Met Office dedicated website: https://ghrsst-pp. metoffice.gov.uk/ostia-website/index.html. Ocean color data are publicly available for download from the ACRI-ST website : https://hermes. acri.fr. Mixed layer depth data are publicly available on the CMEMS website : https://data.marine.copernicus.eu/product/GLOBAL_MULTIYEAR_ PHY_001_030/description.

## Appendix A: Appendix

### A1 Data and Methods

### A1.1 Study sites

The 2023 MHW in the North Atlantic, unprecedented in its extent and intensity, provides a unique opportunity to study the resilience and adaptation of phytoplankton species, including *G.huxleyi*, to extreme temperatures. In the North Atlantic, *G.huxleyi* typically blooms annually in regions situated between the continental shelf of Western Europe (Celtic Sea) and an Arctic shelf Sea (Barents Sea), respectively representing the trailing and leading edges of the bloom distribution (Winter et al., 2014; Neukermans et al., 2018).

The Celtic Sea [14°E - 9°E / 49.7°N - 56°N] is a region where blooms occur annually and MHW resulted in temperature anomalies of up to +5°C in June 2023 (Berthou et al., 2024). The Barents Sea [18°W - 60°W / 68°N - 80°N] is a region experiencing rapid warming and sea-ice loss due to Arctic amplification and "Atlantification" of its water masses (Oziel et al., 2020; Rantanen et al., 2022; He et al., 2024). Within these study sites, a bathymetry mask has been applied to limit turbid waters caused by resuspended bottom sediments and input from rivers, which create false-positive PIC signals. The bathymetric limits are respectively -150 m and -100 m for the Celtic Sea and the Barents Sea and derived from the ETOPO 2022 global relief model at 60 arc-second resolution (MacFerrin et al., 2024).

### A1.2 Satellite data

To assess anomalies in *G.huxleyi* bloom phenology and spatial extent, we used both the daily and the weekly-merged L3 multi-sensor PIC products, derived from MERIS, MODIS, SeaWIFS, VIIRS, and OLCI, providing a 1/24° spatial resolution for 1997-2024 from the GlobColour project (https://hermes.acri.fr/). NASA's standard PIC algorithm (Balch et al., 2005; Gordon and Du, 2001) was used, based on remote sensing reflectance in either two or three bands in the visible and the near-infrared domain (Balch and Mitchell, 2023). Ocean colour observations are limited by the presence of clouds (predominant at high latitudes) which motivate the choice of using weekly-merged rather than daily products for the climatological comparison. To construct a reliable climatology, we employed a 20-year archive (1998-2018), following Cael et al. (2023) who demonstrated that climate change indicators can be derived from ocean color data within a shorter time period than the 30-year WMO recommendation. Daily (weekly) anomalies were calculated by comparing daily (weekly) PIC data to the corresponding constructed seasonal climatology. For PAR, only weekly-merged L3 multi-sensor PAR products are used, derived from MERIS, MODIS, SeaWIFS, VIIRS, and OLCI, providing a 1/24° spatial resolution for 1997-2024 from the GlobColour project (https://hermes.acri.fr/).

For SST, we used, as a reference climatology, the ESA-CCI level 4 Climate Data Record version 3 (CDR, Embury et al., 2024), which offers a daily and globally consistent record at 0.05° spatial resolution. The daily climatology over the 1991-

2020 period is computed with a 5-day moving average. To derive the daily anomalies we compared the daily CDR data to the Operational Sea surface Temperature and sea Ice Analysis (OSTIA) L4 analysis data (Donlon et al., 2012). The SST product is released on a daily basis on a regular latitude-longitude grid with a 0.05° spatial resolution. Marine heatwaves were identified following (Hobday et al., 2016) by comparing daily SSTs with a seasonally varying threshold defined as the local 90th percentile of a 30-year climatology. Periods of at least five consecutive days above this threshold were classified as marine heatwaves.

In the Barents Sea, previous studies have shown that the polar front, separating Atlantic and Arctic water masses, acts as a physical barrier to coccolithophore bloom expansion (Neukermans et al., 2018; Oziel et al., 2020). We therefore computed the Barents Sea polar front position using a local variance filter applied to March-April OSTIA SSTs with a window size of 7x7 pixels (Neukermans et al., 2018). We, first, computed the monthly average from the daily OSTIA archive. Polar Front Waters were then defined as waters having SSTs between the 16th and 84th percentiles (Oziel et al., 2016). The shifting position of the polar front since 1998 is shown in Fig.A6 & A7.

### A1.3  Ocean stratification data

MLD data were obtained from the daily GLORYS12 Version 1 reanalysis, which provides a daily and global record from 1993 to 2024 at 1/12° spatial resolution (Jean-Michel et al., 2021). Vertical temperature and the stratification are derived from the Institute of Atmospheric Physics (IAP) observation-based temperature/salinity fields at 1°x1° horizontal resolution and 41 vertical levels from 1-2000m and a monthly resolution from January 1940 to September 2023 were used. The product is described by (Cheng and Zhu, 2016; Cheng et al., 2017).

Based on this dataset, we define the upper 200-m stratification as the squared buoyancy frequency computed from the density gradient over the top 200-m layer:

$$N^2 = -\frac{g}{\rho}\frac{\partial \sigma_0}{\partial z}\bigg|_{0 \geq z \geq 200}, \tag{A1}$$

where $\sigma_0$ is potential density referenced to the surface, and $g$ is the gravitational acceleration. The squared buoyancy frequency, $N^2$ expressed in $s^{-2}$.

The stratification can be expressed, to a first, approximation, as a linear combination of distinct temperature and salinity contributions(Gill and Niller, 1973):

$$N^2 = N_T{}^2 + N_S{}^2, \text{ with } N_S{}^2 = -g\beta\frac{\partial S}{\partial z}\bigg|_{0 \geq z \geq 200} \text{ and } N_T{}^2 = g\alpha\frac{\partial T}{\partial z}\bigg|_{0 \geq z \geq 200}, \tag{A2}$$

where $\beta$ is the haline contraction coefficient and $\alpha$ is the thermal expansion coefficient.

## A1.4  Bloom detection and phenology

To assess *G.huxleyi* bloom phenology, we applied the methods of Hopkins et al. (2015) on the L3 daily multi-sensor PIC product (Sect.A1.2). This allows us to estimate bloom start and end dates, maximum concentration, and extent knowing the limitations of such data at high latitudes. This method is based on the analysis of the temporal evolution of the PIC concentration over the study site and the identification of both a local minimum before and after the detected peak of the bloom.

The surface extent computation relies on the number of relevant pixel areas detected with a PIC concentration greater than a
region-based threshold (defined on daily products) applied to the weekly-merged L3 products. The threshold is computed based on the 1998-2018 climatology and determined as the PIC concentration on the climatological bloom start date, serving as a baseline for identifying significant anomalies. The respective values for CS and BS are 0.06 mmol.m$^{-3}$ and 0.1 mmol.m$^{-3}$. The study sites are located in mid- and high-latitudes, the surface extent must take into account the surface spherical deformation, defined as follows:

$$S = \sum s_i \text{ with } s_i = 110.574 * \text{latitude} * 111.320 * \text{longitude} * \cos(\text{latitude}) \tag{A3}$$

Based on a global compilation of in situ measurements of coccolithophore species abundance and diversity (O'Brien et al., 2013), the realized ecological niche of *G.huxleyi* (i.e. the environmental conditions under which it can be observed) has been characterized (O'Brien, 2015). Out of seven environmental variables considered, O'Brien et al. (2016) showed that SST, PAR, and MLD were the most important variables influencing coccolithophore diversity. For *G.huxleyi*, the optimal SST range is
situated between 6 and 16°C, optimal PAR between 35 and 42 Einstein.m$^{-2}$.day$^{-1}$, and optimal MLD between 20 and 30 m.

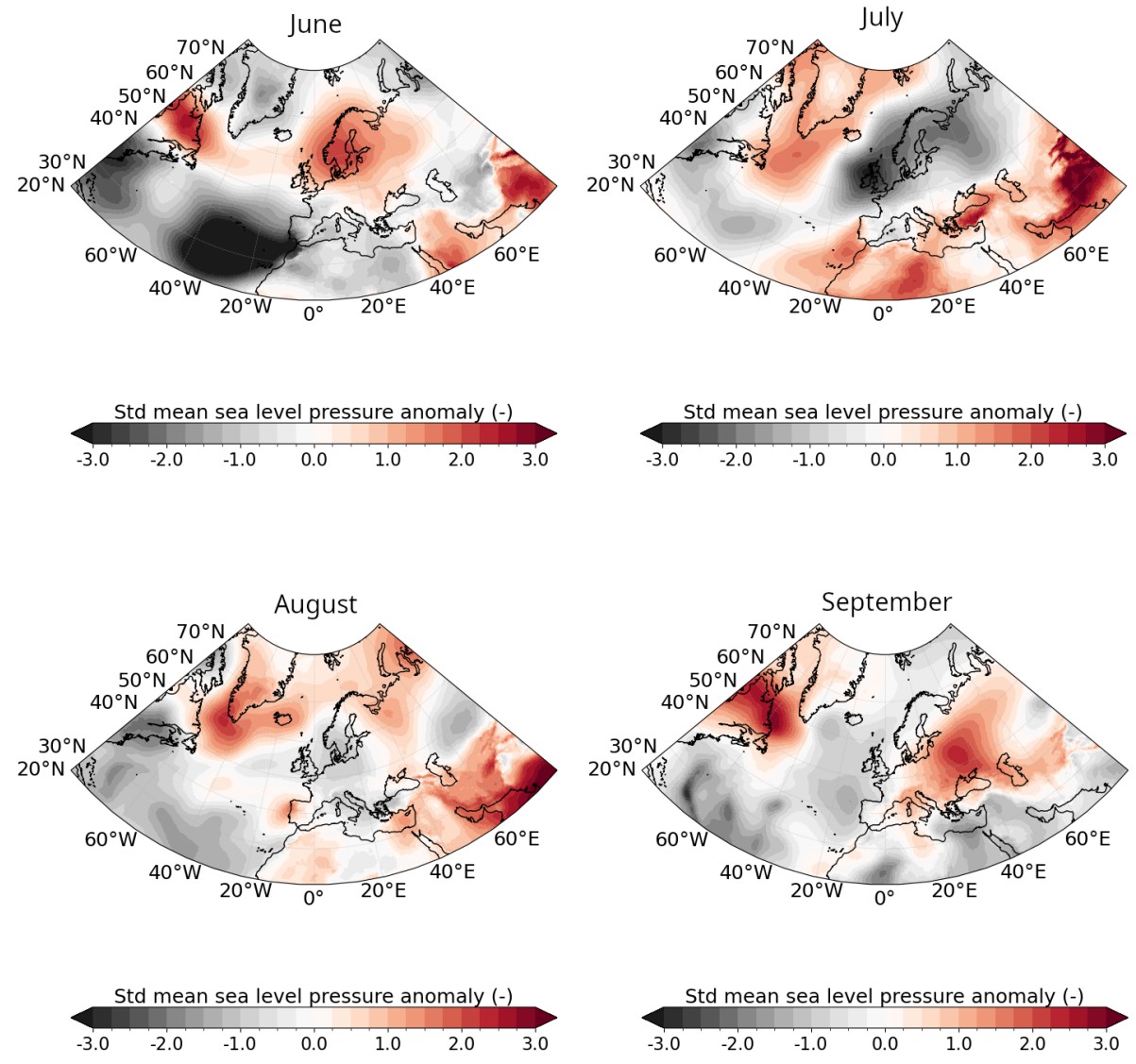

**Figure A1. Ocean-atmosphere conditions in June-July-August-September 2023.** Standardised monthly anomalies from ERA5 in 2023 compared to the 1991-2020 climatological period for mean sea level pressure.

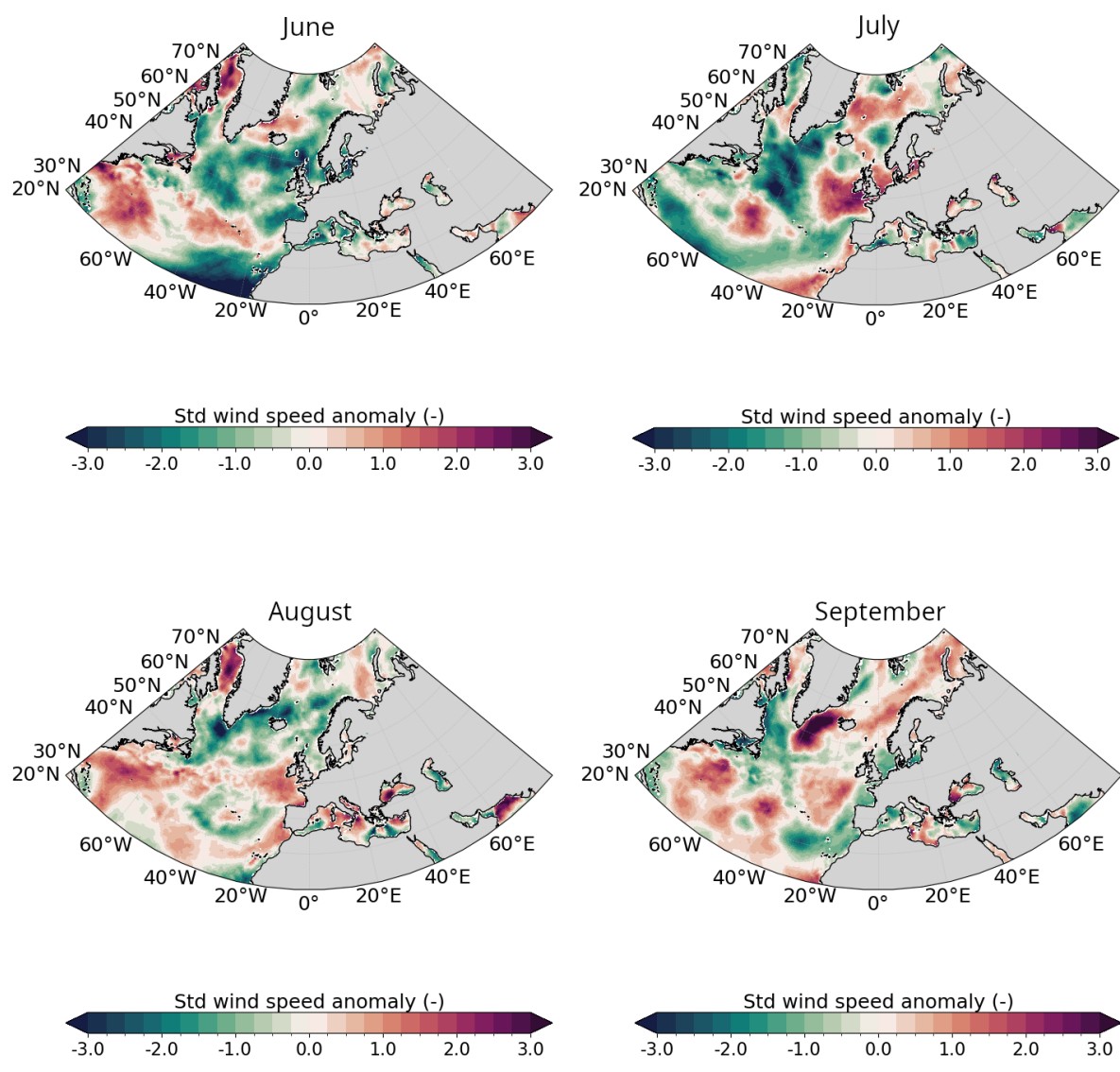

**Figure A2. Same as Fig.A1 for 10-m wind speed.**

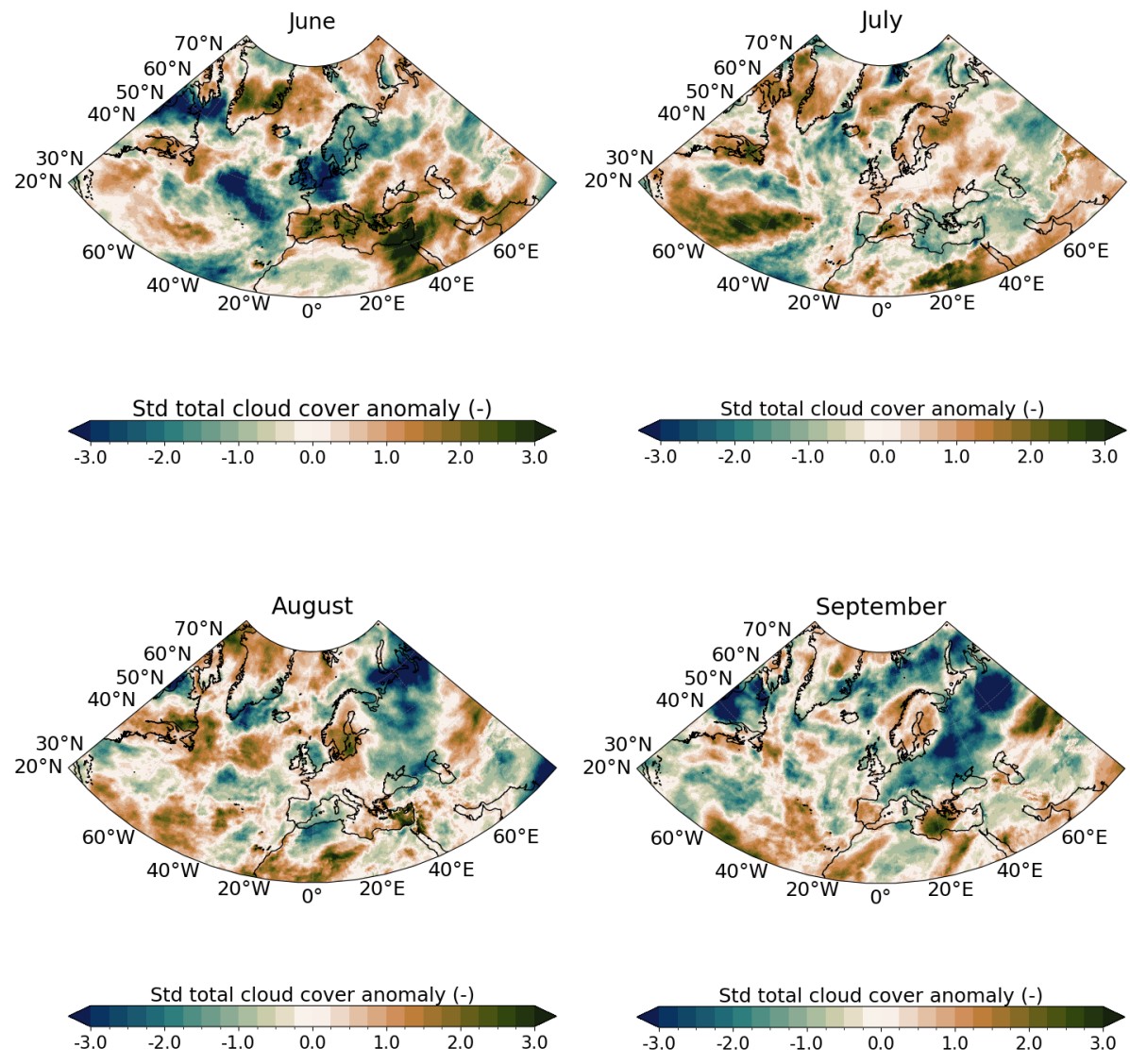

**Figure A3. Same as Fig.A1 for total cloud cover.**

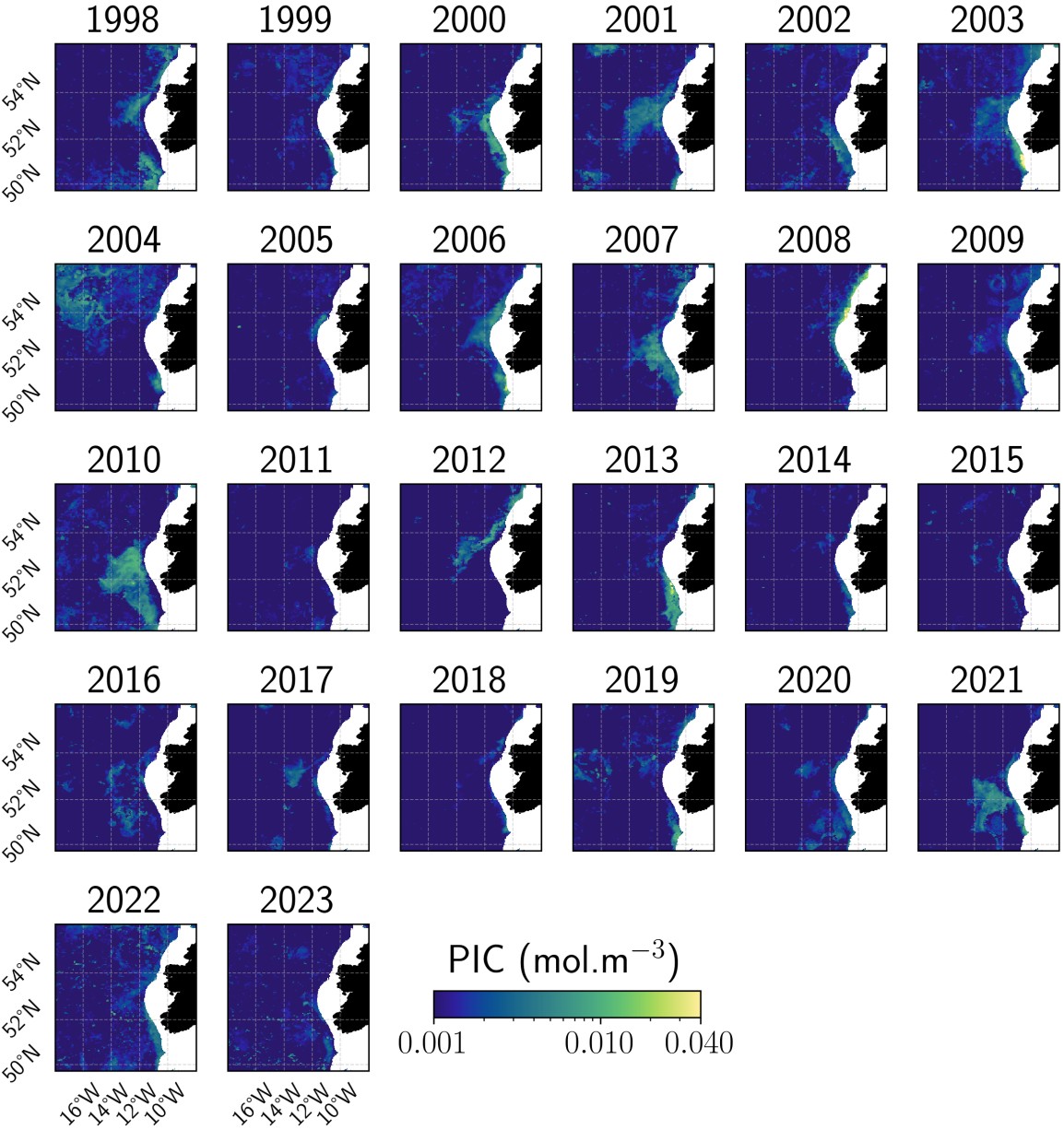

**Figure A4. Summer maximum PIC concentration in the Celtic Sea.** Annual evolution of the remotely-sensed summer maximum PIC concentration. White areas defined coastal zones where the bathymetry is higher than -150m.

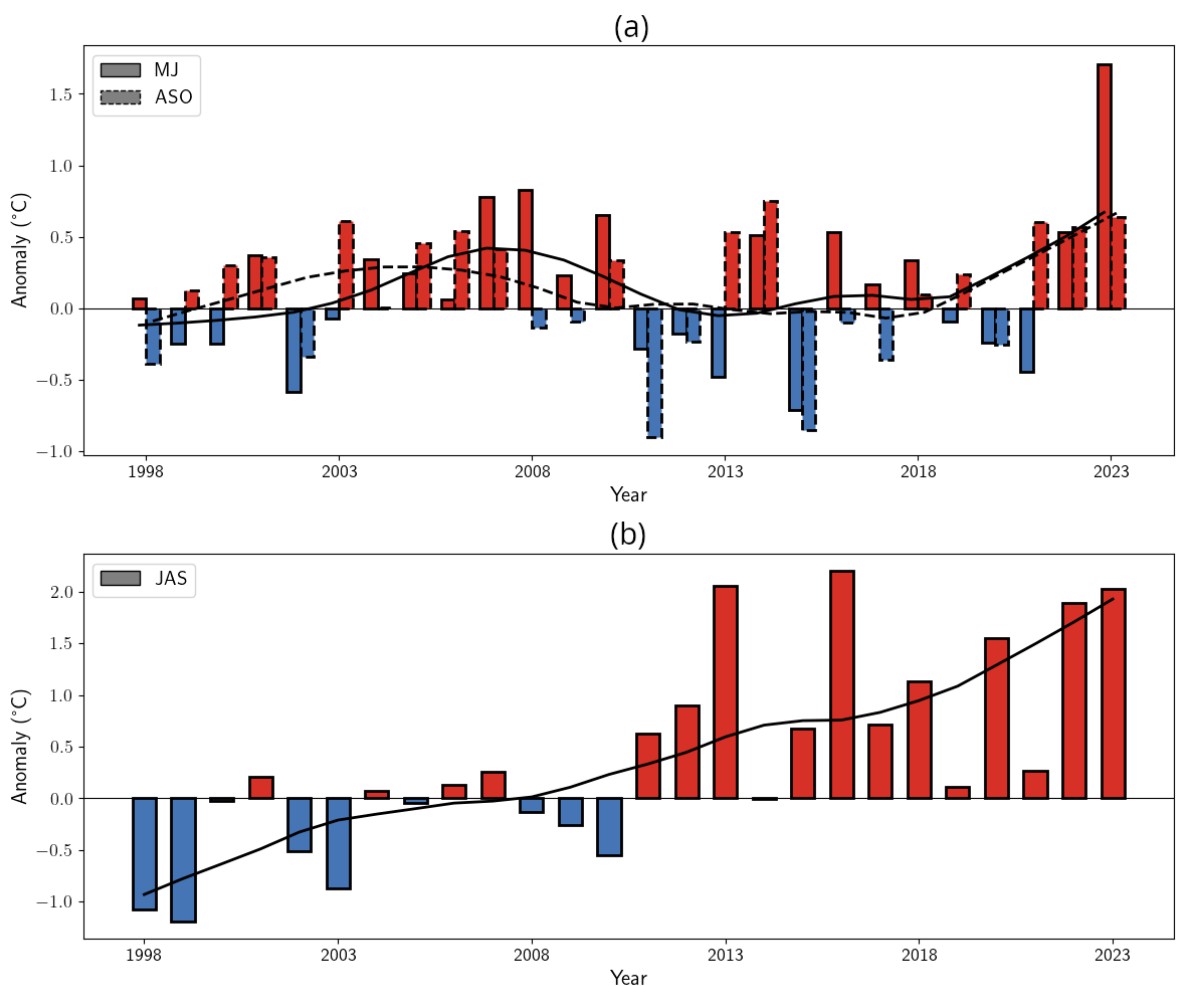

**Figure A5. Long-term SST evolution in the Celtic and Barents seas.** SST anomalies for (a) the Celtic Sea (May-June in solid contour & August-September-August in dashed contour) and (b) the Barents Sea (July-August-September), computed relatively to the 1991–2020 climatological period. Red bars indicate positive anomalies, while blue bars denote negative anomalies. The line indicated the 10-year LOESS trend.

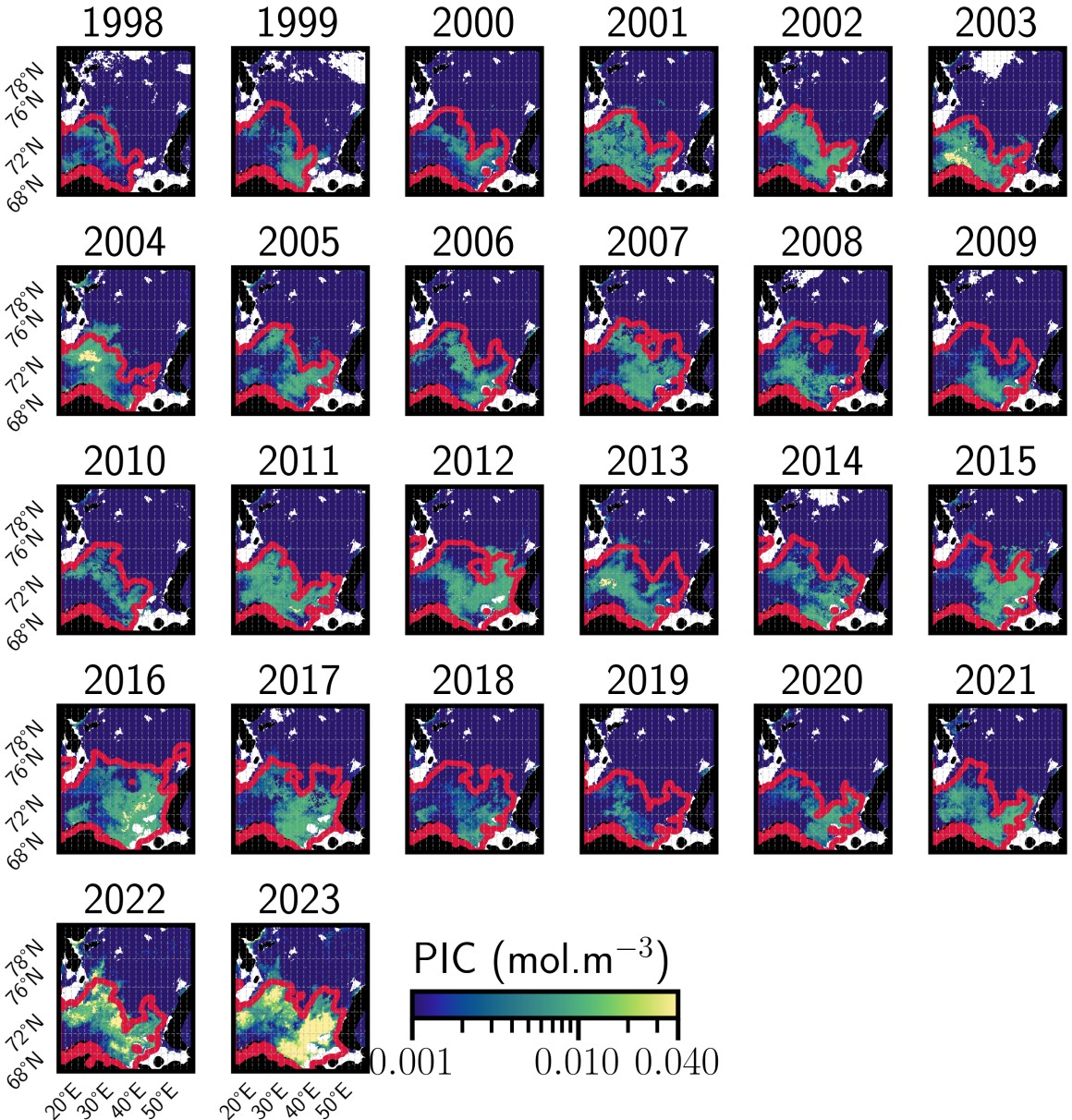

**Figure A6. Summer maximum PIC concentration in the Barents Sea.** Annual evolution of the remotely-sensed summer maximum PIC concentration and the corresponding polar front in red. The polar front is based on an analysis of the ice-free March-April SSTs. White areas defined coastal zones where the bathymetry is higher than -100m.

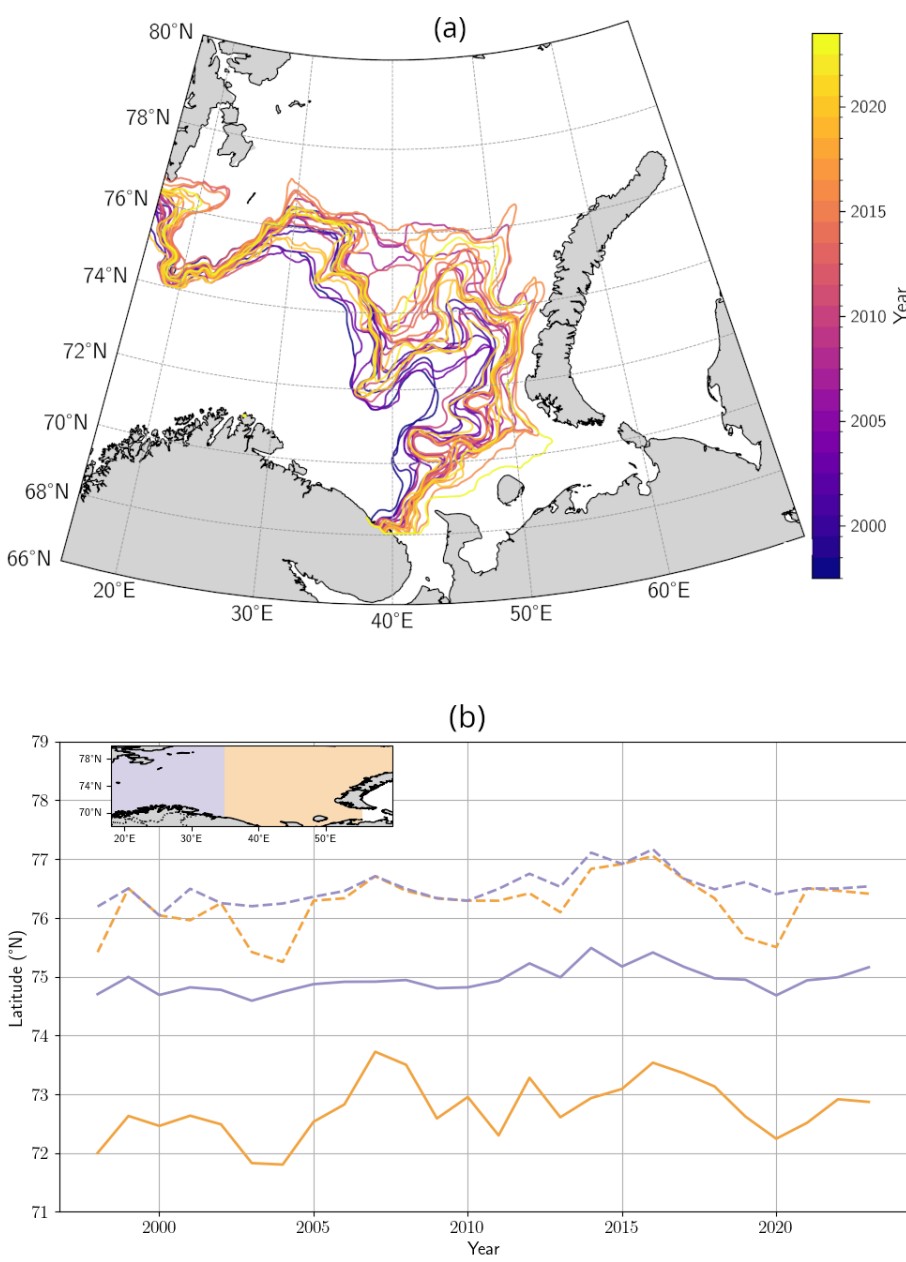

**Figure A7. Shifting position of the polar front in the Barents Sea.** (a) Position of the polar front in the Barents Sea obtained from remotely sensed SST imagery and (b) corresponding position of the polar front maximum latitude in the western (blue lines) and eastern (orange lines) basins of the Barents Sea over the period 1998-2023.

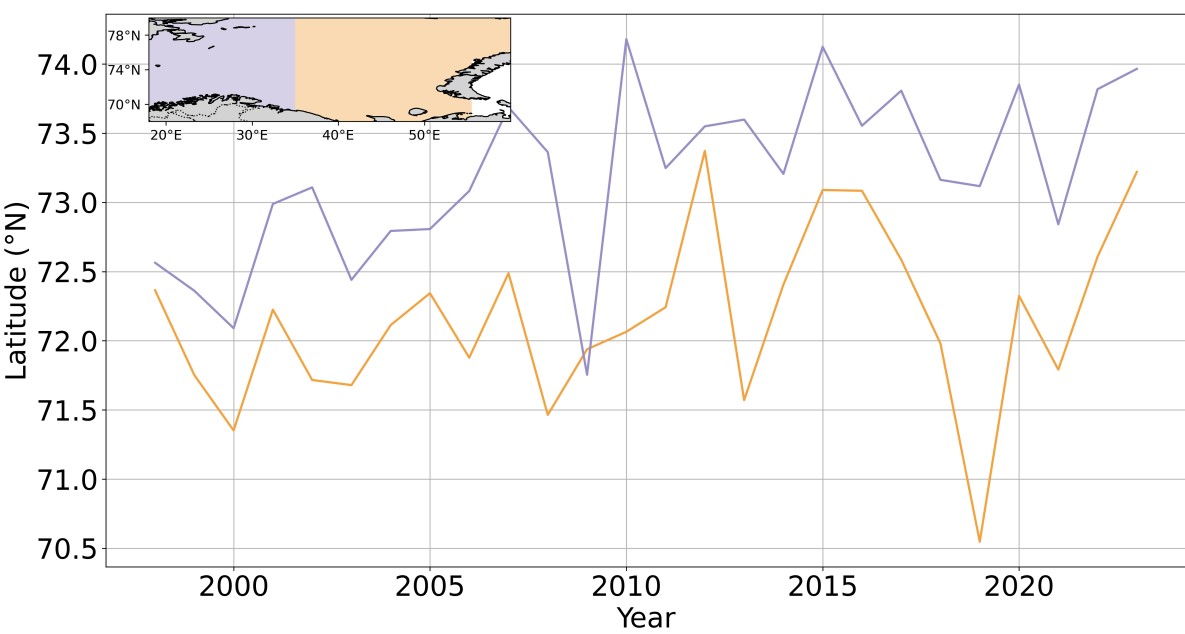

**Figure A8. Shifting position of the leading edge of *G.huxleyi* summer blooms in the Barents Sea.** Temporal evolution of the mean latitude of the bloom summer maximum extent for the western (blue lines) and eastern (orange lines) basins of the Barents Sea over the period 1998-2023.

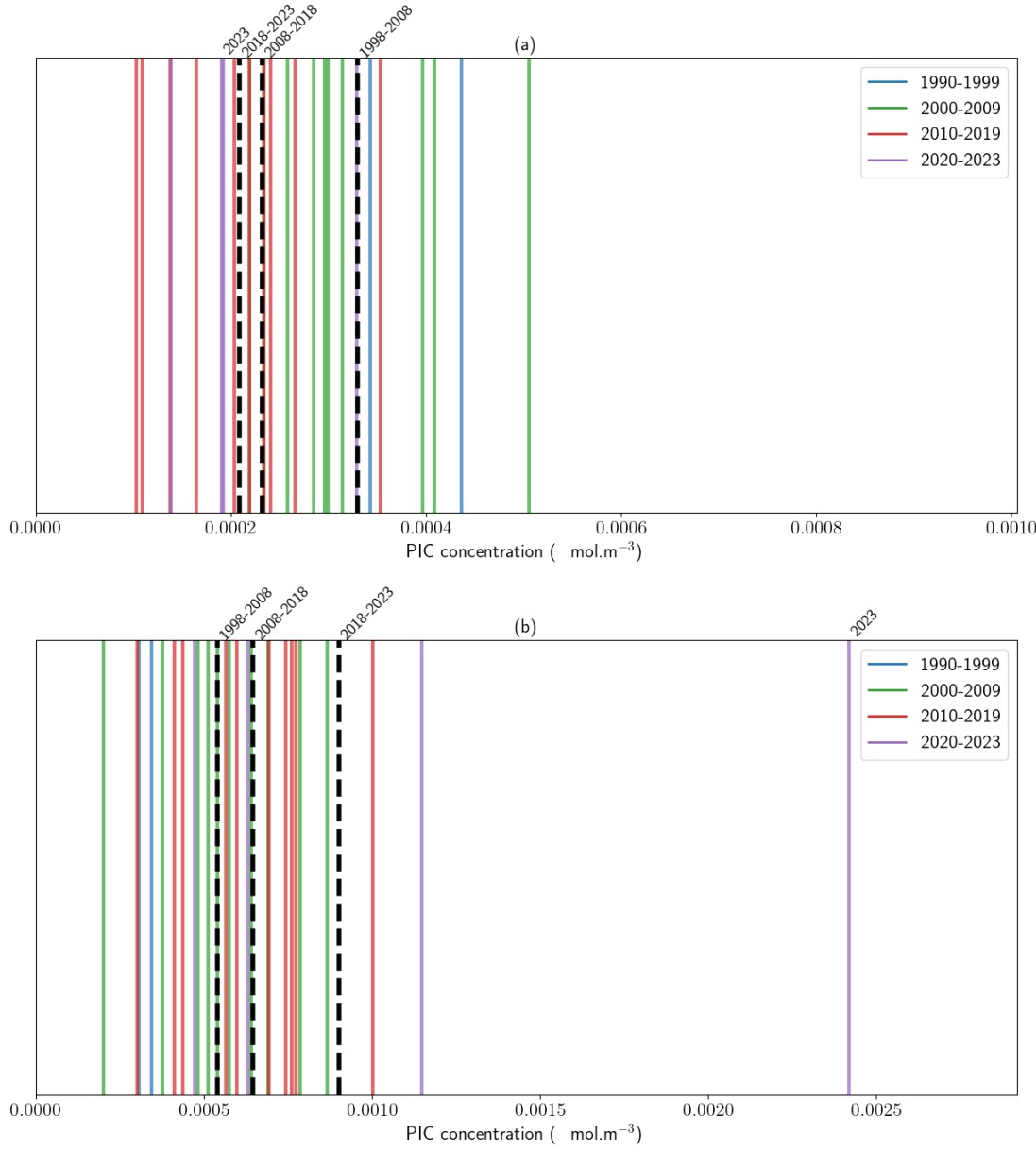

**Figure A9. Decadal evolution of summer mean PIC concentration in the Celtic Sea and Barents seas.** Barcode plots of the distribution of yearly summer (June-July-August) mean PIC concentrations (mol·m$^{-3}$). Colors refer to the corresponding decade with decadal means indicated by black dashed lines. (a) Celtic sea and (b) Barents Sea.

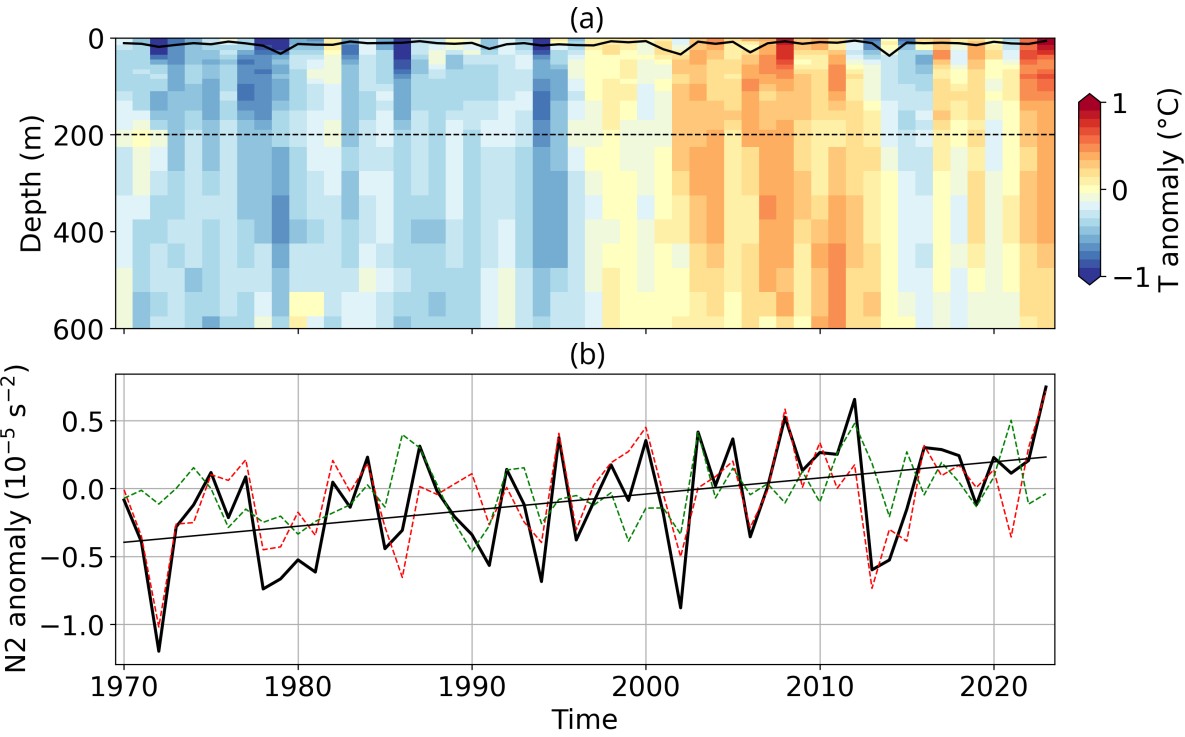

**Figure A10. Stratification conditions in the Celtic Sea.** Vertical profile of (a) May-June anomalous temperature compared to the 1991-2020 climatological mean from the IPA dataset (b) time-series of the 0-200 m stratification anomaly, $N^2$ and temperature (red dashed) and haline contribution (green dashed) to stratification in the Celtic Sea.

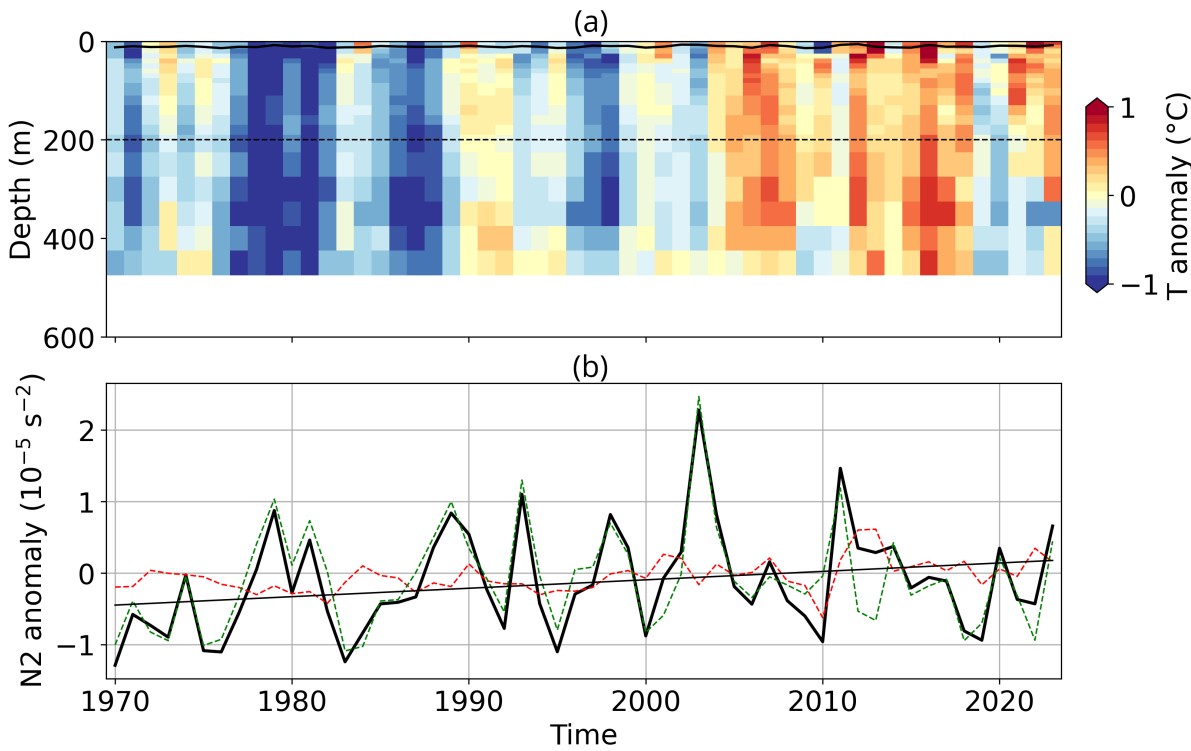

**Figure A11. Stratification conditions in the Barents Sea.** Same as Fig.A10 for the Barents Sea.

*Author contributions.* TG and GN designed the study; TG conducted analysis and wrote the paper with contributions from GN. TG and GN took part in discussions and revisions of the paper.

*Competing interests.* The contact author has declared that none of the authors has any competing interests.

*Disclaimer.* TEXT

*Acknowledgements.* This work is a contribution to CarbOcean (European Research Council under the European Union's Horizon 2020 research and innovation programme Grant agreement No. 853516) awarded to GN. TG sincerely thanks Jean-Baptiste Sallée for many fruitful discussions on the evolution of current dynamics in the Barents Sea and for providing with stratification diagnostics. TG sincerely thanks Christophe Cassou for his mentoring and help in understanding climate internal variability.

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
