# Peer review of "Compounded effects of long-term warming and the exceptional 2023 marine heatwave on North Atlantic coccolithophore bloom dynamics"

_EGUsphere, 2025_

## Author Comment (AC1)

**Reply to reviewer for "Exceptional 2023 marine heatwave reshapes North Atlantic coccolithophore blooms"**

**REV#1**

The authors present a brief descriptive study of the 2023 marine heat waves in the Northeast Atlantic, with two case studies in the Celtic and Barents seas. I found the study to be a decent documentation of this event and I acknowledge the quality of the analysis.

However, I am quite annoyed by the form of the study. Is there a limit to the "fast-science" ? and to the fact of hastily producing a piece of paper with huge or vague statements not specific enough or NOT supported by results ? It's a paper which does not produce any data, new methods or concepts and does not study any process. This is purely descriptive, which would be fine in principle, if the authors would stick to interpreting their results. But this article is greedy and illegitimately intended to be as wide-ranging as possible, mentioning as many buzzwords and fancy concepts as possible (acidification = no analysis, Atlantification = not even defined, poleward expansion = not quantified at all although the authors conclude about a spatial shift.... , ecological niche/hotspots = not defined + not quantified, emergence = not defined/quantified, top-down controls = not analyzed, bottom-up controls = not analyzed, for example, there are interpretations of the role of light without any analysis of light/PAR; or the role of atmospheric conditions with no analysis of atmospheric conditions, sea-ice and ocean currents are mentioned many times also without any result shown).

The fact that they give advice on what people should do next in term of observation and modelling is already annoying, but fine. What is least acceptable is that, in most cases, the claims are not backed up by results. Furthermore, on top of those concepts that are discussed without any analysis, many aspects of the methods are incomplete (see point-by-point comments below: undefined/unprecise concepts: climate variability, Atlantification, optimal bloom development zones ?, trailing/leading regions ? ; incomplete references; missing methods: extent of the bloom ?, leading edge ?, thermal range of coccos?). Many assertions are (most of the time true) but baldly claimed, not even backed up by any references. Due to this bad referencing, the authors presume the reader is an expert (otherwise it's a very complicated read through although it is written quite succinctly) and oblige the reader to look back-and-forth for the needed information throughout the manuscript (many times introduced way after being used: e.g. upper thermal range, never explained).

I also raise a few formal points, i.e. some wording so as not to overstate -sell- the overall importance of E. Huxleyi in the carbon cycle and also in the trophic chain. There is no evidence that they contribute

significantly to the BCP beyond the "ballast effect", as they contribute to a small (~10%) share of the total biomass and the total BCP, release CO2 during calcification, and they are generally avoided by grazers.

Too much is too much, I recommend the authors to take the time to fix this study and to re-submit later on and spare the energy of the readers. A great deal of work is needed before authors can provide analysis to back up all their assertions. An easier alternative would of course be to clarify, or indicate the source of their unsupported results, and/or (re)move them into the discussions.

We would like to thank the reviewer for their comments on our article, for the time they spent reviewing it in detail, and for their help in improving it. The reviewer raised important issues, which we have addressed by adding new figures and analyses to the revised version. We believe that these changes will address his/her concerns. Due to the extent of the work involved, some comments are not addressed point by point, as we decided to revise most sections. In particular, we have added references on the ecological niches of G. huxleyi and proposed an analysis of PAR and polar front trends during the satellite era. This has allowed us to demonstrate the key role of extreme heat events in intensifying blooms.

DETAILED COMMENTS:

METHODS:

Extent of the bloom: I find it not straightforward to understand how the mean/max of the extent of the bloom is derived. Do I understand properly that the bloom start/end is calculated following Hopkins et al. 2015 (with you own custom criterions) pixel by pixel with daily images ? Right ? So the mean/max you are displaying are temporal ? Of daily maps or aggregated monthly maps ? I doubt daily maps have enough coverage to derive a bloom extent, at least in the BS. Also, In which time windows (I guess not the whole year, only the bloom period I guess) ? Maybe try to be more specific in the A1.3.

In this study, we computed regional thresholds based on daily values. We then applied these thresholds to weekly merged products to determine the weekly extent. We applied this method to the entire year, but only weeks within the proliferation period show surface extent.

Sentence in the Methods section :

"The surface extent computation relies on the number of relevant pixel areas detected with a PIC concentration greater than a region-based threshold (defined on daily products) applied to the weekly-merged L3 products."

INTRODUCTION

Lines 25-26 – I thought thin strain has been renamed "Gephyrocapsa huxleyi". Please double-check and eventually fix it. The authors acknowledged the relative small contribution of coccos to the global NPP Line

23 (although I would appreciate some references here). Is the 1-10% contribution is for actually E. huxleyi or for coccolithophores in general? Verify. Same for PIC, orders of magnitudes compared to other carbonated-shell species would help rather than using non-quantitative adjectives such as "important" or "significant" (there are repetitions by the way).

The reviewer is correct that the species "*Emiliania huxleyi*" has been recently renamed "*Gephyrocapsa oceanica*". Thank you for pointing that out; it escaped our attention. We have replaced every occurrence of *Emiliania huxleyi* with *Gephyrocapsa huxleyi* and added the following reference :

Bendif, E. M., Probert, I., Archontikis, O. A., Young, J. R., Beaufort,L., Rickaby, R. E., & Filatov, D. (2023). Rapid diversification underlyingthe global dominance of a cosmopolitan phytoplankton.The ISME Journal, 17, 630–640.

We also modified the introduction to improve precision and added the appropriate reference:

"As photosynthetic organisms, coccolithophores contribute 1-10% to global ocean primary production (Poulton et al., 2007) and about 50% to the deep ocean flux of particulate inorganic carbon (PIC; Neukermans et al., 2023)."

Line 36 & 37 & 60 : cocco or Ehux ? Throughout the manuscript the authors use one or the other inter-changeably. Please stick to one wording consistently.

Within the group of coccolithophores, *G. huxleyi* is the only coccolithophore species that is known to form blooms at temperate to high latitudes. Therefore, in our study, the terms "coccolithophore blooms" and "*G. huxleyi* blooms" can be used interchangeably. We however agree with the reviewer that this can be confusing and have therefore decided to stick to "*G. huxleyi* blooms".

Line 69: space (and coma?) missing:  "(Guinaldo et al., 2025)on top"

Done

Line 72-81: I am a bit puzzled by this paragraph and the use of the term "climate variability" although I see what the authors mean as they refer to Sando et al. 2010. However, the authors of this study in 2010, made sure to re-define this term as they used a pretty narrow definition where climate is limited to ocean heat transport. In ocean modelling, we use climate for atmospheric conditions/forcings for instance. It seems that the authors here are making the confusion (or just are not being specific enough in the wording, maybe replace "climate variability" by precisely what you actually mean OR define it) between atmospheric conditions, climate variability and ocean heat transport. It seems that the authors are trying to explain that warming in winter is a remote effect (through advection – ocean heat flux) whereas in summer, it's a local effect (local atmospheric warming). Could you re-phrase this paragraph?

We were mentioning climate variability at various timescale, e.g. multi-decennal variability related to the AMV, interannual variability related to the atmospheric conditions/weather regimes. However, we agree that our mention of "climate variability" is indeed confusing. Therefore, in the revised version, we decided to modify these lines.

Sentence modified :

"Even at the northern edge of the North Atlantic, the BS atmospheric and oceanic internal variability responds to NAO conditions (Levitus et al., 2009; Chafik et al., 2015), while summer conditions favor the likelihood of high-pressure blocking systems over northern Europe (Rantanen et al., 2022; Rousi et al., 2022) characterized by weak winds and high solar radiation."

Line 84: thermal range of coccos, which is ? there is a tendency to build sentences like this one with "while" in middle connection two informations which are not related, quite confusing to read and energy-demanding to reconstruct. Maybe you mean something like :

" In the CS, oceanic conditions remained favorable for coccolithophores until mid-June … while … the second half of summer was marked by less favorable conditions. "

Furthermore, I do not understand how temperatures exceeding upper thermal range can be a favourable condition for growth. If I try to the brain gym, I go in the first paragraph (or Fig. 2b) and I infer SST is about 17.5degC in the CS. What is the upper thermal range definition ? It is not define unless I missed the obvious. Help the reader, repeat numbers, define concepts, and repeat references to figures (Fig. 2a,b), etc.

We agree that there is a lack of detail regarding the temperature range and other specific aspects of the phenology of G. Huxleyi. In the revised version, we have included references supporting this temperature range and details in the main text and in the section on methods.

Main text :

"To evaluate the impact of MHW on *G. huxleyi* blooms, we examine impacts on the three most influential environmental variables that characterize the ecological niches of coccolithophore species, namely SST, Photosynthetically Active Radiation (PAR), and the depth of the mixed layer (MLD) (see Sect.A1.4; O'Brien et al., 2016). For *G. huxleyi*, the optimal SST range was found to be situated between 6 and 16°C, optimal PAR between 35 and 42 Einstein.m−2.day−1, and optimal MLD between 20 and 30 m (O'Brien, 2015). These ranges were extracted from the realized ecological niche of *G. huxleyi* (i.e. the environmental conditions under which it can be observed) set up by O'Brien (2015), based on a global compilation of in situ measurements of coccolithophore species abundance and diversity (O'Brien et al., 2013)"

Methods section :

"Based on a global compilation of in situ measurements of coccolithophore species abundance and diversity (O'Brien et al., 2013), the realized ecological niche of *G. huxleyi* (i.e. the environmental conditions under which it can be observed) has been characterized (O'Brien, 2015). Out of seven environmental variables considered, O'Brien et al. (2016) showed that SST, PAR, and MLD were the most important variables influencing coccolithophore diversity. For *G. huxleyi*, the optimal SST range is situated between 6 and 16°C, optimal PAR between 35 and 42 Einstein.m−2.day−1, and optimal MLD between 20 and 30 m (O'Brien, 2015)"

Lines 99: "The primary limiting factor for blooms in the BS is the photosynthetically active radiation (PAR), which drives bloom onset and decline."

That is a bold statement without any reference. Maybe true in general, but we know that Ehux occupy a niche that is not only driven by light. It is for example shown that viral lysis can terminate such blooms. Plus, this is interpretation because your findings do not back up this result.

Additionally, you start the paragraph with "The PIC dynamics reflected these environmental conditions" and finishes with "This bloom dynamic correlated with the environmental

Conditions" . vague statement, never quantitative or specific. No description of environmental description is given in this paragraph so the reader has to remind perfectly the previous section. Painful. Try to be specific, the earlier the better. What bloom dynamic (peak? Bloom duration? )

To account for both the comment of Rev#1 and Rev#2 the section 2.1 has been revised accordingly with the inclusion of a specific paragraph dedicated to PAR analysis and atmospheric conditions.

"Likewise, PAR in CS was strong in May-June with values surpassing 42 Einstein.m−2.day−1 (upper-range of the optimal conditions for G.huxleyi with thresholds established from the study of the species' realized ecological niche; see Sect.A1.4 and O'Brien (2015)) with conditions becoming more favorable in July onward (Fig.2c). These variations are primarily influenced by the atmospheric conditions, specifically cloud cover. In June, a persistent high-pressure system over Fennoscandia (Fig.A1) led to exceptionally weak wind conditions (Fig.A2) and low cloud cover (Fig.A3) but increased toward climatological values onward . In BS, PAR was exceptionally high compared to the summer climatology allowing sufficient sunlight to reach the surface ocean for photosynthesis throughout summer (Fig.2d). These results are influenced by the cloud cover over BS where a large portion of the sea experienced significant clear-sky conditions during summer (Fig.A3)"

We also added figures to analyse the environmental conditions (Fig A1, A2 & A3) and the PAR evolution in 2023 (Fig 2 c-d).

**Figure 1. Ocean-atmosphere conditions in June-July-August-September 2023.** Standardised monthly anomalies from ERA5 in 2023 compared to the 1991-2020 climatological period for mean sea level pressure

**Figure A2. Same as Fig.A1 for 10-m wind speed**

**Figure A3. Same as Fig.A1 for total cloud cover**

**Figure 2. Daily spatially averaged SST, MLD and PAR variables for 2023.** Spatially averaged SST (black solid line) and MLD (blue solid line) anomalies for 2023 in (a) the Celtic Sea and (b) the Barents Sea. The black dashed line represents the climatological SST averaged over each basin for the period 1991–2020, while the green dashed line marks the 90th percentile threshold for MHWs, as defined by (Hobday et al., 2016). Red shading indicates periods of MHWs, while grey shading highlights conditions favorable to G.Huxleyi blooms based on optimal ranges for SST and MLD in the species' realized ecological niche (see Sect.A1.4; O'Brien, 2015). Spatially averaged PAR for 2023 in (c) the Celtic Sea and (d) the Barents Sea. The vertical brown lines inform on the optimal range for G.Huxleyi blooms (see Sect.A1.4)

Line 102: "potentially reflecting the multi-annual North Atlantic variability…"

Another interpretation that should go in the discussion and that is not supported by results here. Speculation.

This sentence and the related notion are not a result. We have decided to remove it.

Line 109: "linked to increased atlantification" where /how do you document Atlantification ? By the position of the polar front ? Say it here. This article is for experts only. You say it later lines 115-116.

We agree on the lack of detail regarding the position of the polar front and its consequences on PIC distribution. We have added two figures (A7 and A8) and relevant references to support our findings.

**Figure A7. Shifting position of the polar front in the Barents Sea.** (a) Position of the polar front in the Barents Sea obtained from remotely sensed SST imagery and (b) corresponding position of the polar front maximum latitude in the western (blue lines) and eastern (orange lines) basins of the Barents Sea over the period 1998-2023.

**Figure A8. Shifting position of the leading edge of G. huxleyi summer blooms in the Barents Sea.** Temporal evolution of the mean latitude of the bloom summer maximum extent for the western (blue lines) and eastern (orange lines) basins of the Barents Sea over the period 1998-2023.

Line 116: "Interannual variability in the position of the polar front is accompanied by shifts in PIC maxima," another vague and non-demonstrated statement. Is there a cause-consequence demonstration that the position of the polar front influence PIC max ? I don't see it at least.

References are provided in the study to demonstrate this point (Oziel et al. 2020, Neukermans et al 2018). We've also added Fig A7 (see comment above) about the polar front shifting position and corresponding shifting position of PIC max (Fig A8, see comment above) with also a mention in the Results section.

Line 122: First reference of methods here as section A1. Which should actually be A1.3. If you want to use methods as supplementary, you need to be irreproachable. You are not. References to other methods sections are not proper. All methods should be cited like here in order: study cite when you first introduce BS and CS, Satellite data. No MHV definition. Btw, MLD is no satellite data.

We agree with the reviewer that our references to methodological sections lacked accuracy. . We have carefully checked the entire manuscript on these aspects (order of the methods and their referencing) and corrected where necessary.

Line 145: double bracket ((. The studied area is the Arctic ? I thought it was the North Atlantic? Barents Sea could be both as a frontal area but you need, again, to be consistent. Sea-ice melt induced stratification does not "facilitate the accumulation of nutrients". It just stratifies. And then "These processes likely contributed to the unprecedented bloom intensities observed in recent years." … A purely speculative paragraph in the result sections…

We agree on the speculative nature of the paragraph and have removed it, as we did for the sentence on sea-ice melt and stratification.

Line 152: "Here, the bloom period remains limited by PAR availability" Where is the demonstration ??? Are we doing science here ?

Our revised analyses now include observational evidence on PAR availability (Figure 2 c and d, see comment above).

Line 157: So now "The establishment of these temperatures was locally modulated by climate variability" so what is meant here ? Climate variability is ocean heat transport or atmospheric warming. If the former, it's not local, it's remote. If you mean atmospheric warming, then you have revise entirely the introduction and better frame/define.

In this paragraph, we addressed variability at different timescales and its impact on SSTs. This sentence refers to the amplification of the 2023 MHW event by shorter-scale atmospheric variability and multidecadal variability (AMV) and the related reference. Recognizing that this sentence was not necessary in the context of this study, we decided to delete it.

Line 160: Coccos or E. Huxleyi ?

In line with a previous comment about consistency, we decided to stick to *G. huxleyi*.

Line 161: Another new un-defined term: "optimal bloom development zones"

We acknowledge that we used different terms for the same concept. We therefore checked for consistency throughout the document. This term was part of a sentence that was deleted in the revised version.

Line 162: "in trailing/leading regions" not defined or referenced to methods. The authors are asking the readers to read their mind. Definition in methods is incomplete.

Trailing and leading edge are now defined and associated with references in the Methods and the main text.

Main text :

"the Celtic Sea and the Barents Sea, respectively representing the trailing (or equatorward) edge and the leading (or poleward) edge of G.huxleyi bloom distribution in the North Atlantic Ocean (Winter et al. 2014)"

Methods :

"In the North Atlantic, G. huxleyi typically blooms annually in regions situated between the continental shelf of Western Europe (Celtic Sea) and an Arctic shelf Sea (Barents Sea), respectively representing the trailing and leading edges of the bloom distribution (Winter et al, 2014, Neukermans et al, 2018)"

Line 164: Okay, but you do not investigate any ocean currents… am I right ? Why ?

Yes, the reviewer is right that ocean currents were not specifically investigated - the polar front position was. Therefore, we have removed this statement.

Line 166: Impact the surface area ? What does that mean ? it changes the surface area of the Barents Sea ?

We acknowledge that this sentence lacks clarity. We have revised the conclusions section entirely, which no longer includes this statement.

Line 175: and the fact that coccolith sheds light when they shed, i.e. when the bloom is dying…

It is unclear to us what this comment refers to.

Line 176: or modelling them?

This study intentionally focuses on an observational analysis of the impact of MHW on PIC. We fully understand the need for modeling to disentangle the various factors and quantify the extent to which each contributes to blooms, but we consider this to be beyond the scope of our study.

Line 178: Now another concept: ecological niche. First time. Not defined, not characterized.

Details added in the Methods section:

"Based on a global compilation of in situ measurements of coccolithophore species abundance and diversity (O'Brien et al 2013), the realized ecological niche of G. huxleyi (i.e. the environmental conditions under which it can be observed) has been characterized (O'Brien et al 2015)."

Line 180-182: Did I see an analysis on atmospheric winds ? Stroms ? Air temperature ?  PAR?

We agree that this is a major weakness. We have added figures A1, A2 & A3 showing atmospheric conditions from May to September and the temporal evolution of PAR in both regions.

Line 190 : This lesson is hard to take by a study which does not produce any data or study any process.

Rephrased as follows:

"This effort could be developed by considering a combination of multi-scale observation networks capable of providing the initial conditions, and enhanced modelling frameworks that capture subsurface dynamics and multistressor
interactions to anticipate future changes and inform adaptive strategies for marine ecosystems (Gregg and Casey, 2007; Nissen et al., 2018; Krumhardt et al., 2019)."

Line 195: "This study reaffirms the poleward expansion of temperate phytoplankton communities" This is a lie … where is documented this poleward expansion ? You did not bring any analysis that support that or add to the previous literature. Same for "highlights the emergence of new ecological hotspots in high-latitude regions". Words mean something. Have you conducted a time-of-emergence analysis ? How did you show an emergence ? How do you support it ?

This section has been completely revised to meet both Rev#1 and #Rev2 comment. Specifically, we have added an analysis of the shifting polar front position (Figure A7, see comment above) in the revised manuscript) demonstrating the poleward expansion of G. huxleyi blooms.

Line 198: "These shifts, while globally evident, impact regional biogeochemical cycles and food web dynamics." I do not understand the sense of this sentence, what is evident at global scale ? And How does this oppose regional bgc cycles and food web dynamics.

This section has been completely revised to meet both Rev#1 and #Rev2 comment.

Line 199: "Predatory species" … un-related. Coccos are not an important food source, neither a big share of the phyto biomass.

It has indeed long been thought that coccolithophores are not an important food source for zooplankton. However, recent work by Dean et al. (2024 https://www.science.org/doi/10.1126/sciadv.adr5453 ) or Meyers et al. (2020, https://doi.org/10.1016/j.pocean.2018.02.024 ) shows that microzooplankton can exert strong top-down control on both bloom and non-bloom coccolithophore populations, grazing over 60% of daily growth. Furthermore, microzooplankton grazing is now considered a major driver of the dissolution of calcite in shallow waters (e.g., Dear et al. 2024, Ziveri et al. 2023; https://www.nature.com/articles/s41467-023-36177-w )

Line 204: No reference for arctic acidification ? Really.

The reference was not displayed in the first version. Corrections made.

Line 207 : It is not counter-intuitive, sea-ice is melting away (even in winter) and there is more ocean-atm interactions/forcing. Plus there is a compensation effect with the outflowing freshwaters in the Fram strait. So this is both a buoyancy and mechanical effect.

Sentence removed as it is beyond the scope of the study.

Line 209: wrong placement of citation. Sallée et al. 2021 is about MLD and stratification ONLY.

Correction made.

"These dynamics, including the vertical variation of the summertime mixed-layer depth (Sallée et al., 2021), may reduce both light and nutrient availability, and also have implications for carbon export, a critical function of calcifying species"

Line 209: speculations again and again. How is that calcicfying species is a critical carbon sequester ? I though calcification produces CO2 ? I am teasing because you just through out concept without explaining anything. The carbonate pump of course is responsible for a small share of the BCP. But why this is important them ? Through What process ? What is sequestration ? Do you define it ? it's not trivial at all.

Coccolithophores play a complex role in the carbon cycle as on the one hand their calcification produces $CO_2$, but on the other hand, their dense calcite scales are thought to enhance the sinking of organic matter to the deep ocean through the so-called ballast effect of aggregates. This has been a topic of debate for over two decades (as discussed in detail in Neukermans et al. 2023, Earth Science Reviews).

This dual role has been referred to in the introduction and has been reiterated only very briefly in the conclusion section as follows: "Additionally, the evolution of water column stratification plays a key role in promoting blooms with a clear signal in the North Atlantic which in fine may alter the regional carbon cycle."

Line 212: "the Barents Sea's historical increases in bloom intensity may reflect enhanced nutrient inputs, favorable light conditions, and prolonged ice-free seasons driven by Arctic warming." Coccos do not need much nutrients, check literature. Are coccos in the Arctic Waters ? I though they were staying south of the polar front. What connection with sea-ice then ? For the BS, how can you discard grazing pressure ? Viral lysis ? other losses ? Which driver is more important ? How do you choose what is important ? Also for CS, I mean, is there less light in the CS ? Less nutrients? Do you provide support for any claim ?

This section has been completely revised to address comments Rev#1 and #Rev2. In the new Conclusion section we have removed the sentence.

Line 215: "Tipping point" It feels like the authors need to name drop every fancy concepts.

We acknowledge the sentence does not provide enough information to discuss the results and we have decided to remove it.

Line 216: First appearance of modelling…

While designing the study we made the choice to rely on satellite observations and describe the consequences of MHW. We acknowledge this may introduce biases but are convinced about the potential of observations only to evaluate the impacts.

Line 242: trailing edge / leading edge of what ? the blooms or the North Atlantic… ?

As mentioned before, we added details on trailing and leading edge of bloom distribution.

"In the North Atlantic, G. huxleyi typically bloom annually in regions situated between the continental shelf of Western Europe (Celtic Sea) and an Arctic shelf Sea (Barents Sea), respectively representing the trailing and leading edges of the bloom distribution (Winter et al, 2014, Neukermans et al, 2018)"

Line 245: provide ETOPO version.

Version added : "The bathymetric limits are respectively -150 m and -100 m for the Celtic Sea and the Barents Sea and derived from the ETOPO 2022 global relief model at 60 arc-second resolution (MacFerrin et al., 2024)"

Line 278: Ah ! So the upper thermal range is here, and is 16degC ? How those criterions have been decided ? Is it arbitrary ?

References added to explain how these ranges have been decided.

"Based on a global compilation of in situ measurements of coccolithophore species abundance and diversity (O'Brien et al 2013), the realized ecological niche of G. huxleyi (i.e. the environmental conditions under which it can be observed) has been characterized (O'Brien et al 2015)."

---

## Author Comment (AC2)

**Reply to reviewer for "Exceptional 2023 marine heatwave reshapes North Atlantic coccolithophore blooms"**

**REV#2**

The manuscript «Exceptional 2023 marine heat wave reshapes North Atlantic coccolithophore blooms» assesses the impact of the 2023 big North Atlantic marine heatwave on blooms of a key coccolithophore species in terms of bloom intensity, extent, and phenology in both the Celtic Sea and the Barents Sea. The study finds a decline in bloom intensity and extent in the Celtic Sea, while the Barents Sea experiences record-breaking bloom expansion, which, according to the authors, is likely due to ongoing Atlantification and sea ice retreat. The authors discuss the implications for the carbon cycle and marine food webs, emphasizing the importance of continued monitoring in the context of current climate change.

I found the study to be scientifically relevant as it addresses the response of fundamental marine organisms to extreme climate events. Especially, as coccolithophore plays a role on carbon sequestration and on climate regulation through the production of DMS. However, I found this article to be chiefly descriptive, and to not contribute any new data or new methods. I think it is reviewing and commenting concepts that are not all provided by their results. It does not directly examined the mechanisms underlying the observed bloom changes, but rather relies on other studies. The study frequently references broad concepts such as Atlantification, ecological niches, or acidification yet lacking of specific information (nor even concise definitions) or supporting analyses. These terms are (look like) used as buzzwords without being their effects quantified, and with insufficient evidence nor support by the article results. I also believe the paper is not written for non-specialists, as the referencing is quite incomplete and some key concepts are not explained at all.

Though I think the use of satellite data is justified and the variables selected are appropriately used, some methodological definitions such as how bloom extent and leading/trailing edges are defined and calculated, or what is the thermal range for E. huxleyi blooms, are missing.

Finally, I think that authors overstates the role attributed to the coccolithophore species in the carbon cycle. Authors suggest a major contribution to the biological carbon pump, but not providing sufficient evidence for that.

All in all, I think that the structure of this paper need to be revised in order to provide justification for the observed changes in the coccolithophore blooms, and including analyses of their potential drivers as, by now, it is unclear which novelties are specifically provided by their own results.

We would like to thank the reviewer for their comments on our article, for the time they spent reviewing it in detail, and for their help in improving it. The reviewer raised important issues, which we have addressed by adding new figures and analyses to the revised version. We believe that these changes will address his/her concerns. Due to the extent of the work involved, some comments are not addressed point by point, as we decided to revise most sections. In particular, we have added references on the ecological niches of G. huxleyi and proposed an analysis of PAR and polar front trends during the satellite era. This has allowed us to demonstrate the key role of extreme heat events in intensifying blooms.

Specific comments (I just provided few comments as I think the manuscript needs a thorough revision)

Lines 16 and 21: Here, which conditions are specifically referred to.

Sentence modified :

"During boreal spring and summer, large parts of the North Atlantic Ocean are transformed into shades of color, indicating the occurrence of phytoplankton blooms."

Lines 35 to 40: It is not clear if author claim that coccolithophore are affected by or can resist acidification and warming.

Sentence changed to "the inhibitory effects of ocean acidification may limit coccolithophore calcification in the Arctic, despite the region's rapid warming (Schlüter et al., 2014, Smith et al., 2017)."

Line 43 and 52 (where it is redundantly written): Are authors claiming that long-term warming and internal variability the drivers of MHWs?

MHWs are primarily driven by oceanic and atmospheric processes that are part of internal variability (synoptic conditions leading to increased solar radiation, below-average cloud cover, reduced winds, and turbulent mixing), as demonstrated in the review article by Holbrook et al. 2020. In addition to these processes, extreme events are amplified by long-term warming, which affects both the baseline and stratification, as shown in recent studies (Guinaldo et al. 2025, England et al 2023, Gyuleva et al. 2025).

However, in the revised version we removed this sentence.

Line 47: What "hazards arising from different sources.."?

Sentence changed to : "These consequences are exacerbated by a combination of biogeochemical or atmospheric known as compound events (Zscheischler et al., 2018; Burger et al., 2022; Le Grix et al., 2022)"

Line 55: In June, but for how long?

Mention of 16 days added :

"In 2023, a record-breaking marine heatwave developed, resulting in SST anomalies exceeding +5°C across broad areas of the shelf for 16 days in June"

Lines 99 and 103: If these are the main drivers of the coccolithophore blooms duration and extension, then what are (quantification) the role of MHWs?

In the revised version, we have modified the phrasing and shown that the evolution of the polar front alone cannot explain the intense blooms of 2023 (see Fig A7 and A8) and that record high summer SST are highly correlated to the bloom extent (Table A1) in addition to PAR intensity (Fig 2c-d).

However, quantification is limited in an observation-based analysis and must take into account modeling/attribution studies that are beyond the scope of this study.

Lines 118 to 120: I think this is a vague sentence without referenced.

This section has been modified and we added further analysis.

Instead of :

"However, in BS 115 (Fig. A3), a strong northeastward shift in summer maximum concentrations was observed, aligning with the shifting position of the polar front and thus the atlantification of the water masses. Interannual variability in the position of the polar front is accompanied by shifts in PIC maxima, likely driven by bio-advection processes transporting particulate material along the front (Oziel et al., 2020). Years 2004, 2010, and 2023 exhibited larger areas of elevated PIC (Fig. A3). This underscores the compound effect of the Atlantification and ocean warming on the shift of optimal conditions and the enhancement of such a 120 situation under extreme MHWs events like in 2023."

we propose :

"In contrast, the BS exhibited a northeastward shift in summer maximum concentrations (Fig.A6 & A8). While the western BS shows limited front variability and no consistent trend, the eastern BS is characterized by high interannual variability and a long-term northward shift of 300 km for the northernmost position of the bloom and a shift of 155 km for the latitudinal mean position of the bloom. Even though both the latitudinal mean front position have regressed since 2016, another level close to the record high was reached in 2023 (Fig.A7b), exhibiting a spike in the northward maximal expansion in 2022 and 2023 (Fig.A6). This spatial reorganization of plankton distribution in the Barents Sea has been associated with 'Atlantification', which in turn enhances blooms of temperate phytoplankton such as G. huxleyi through bio-advection (Oziel et al., 2017). However, this phenomenon does not fully explain the exceptional bloom

observed in 2023 even if the interannual variability in the position of the polar front is accompanied by shifts in PIC maxima (e.g. 2004, 2010; Fig.A6 & A7 & A9)."

Line 178: Then, why 2023 is different to other years?

Sentence modified by :

"The changes observed in 2023 and reaching exceptional level are an extreme signature of multi-annual variability superimposed on long-term trends."

Line 195 to 215: This paragraph seems to discuss other studies but not including properly the results of the present manuscript.

As proposed by Rev#2, we have completely revised this section by moving some paragraphs and writing new ones. Instead of discussing other studies, the aim is to highlight the main limitations and gaps in this study and the additional analyses needed to achieve a comprehensive understanding of the impacts of MHWs on coccolithophores

"Coccolithophores, like other calcifying organisms, are sensitive to ocean acidification, potentially reducing their ability to produce coccoliths. Polar regions, subject to increased ocean acidification (Gattuso and Hansson, 2011), may become less favorable for these organisms in the long-term (Terhaar et al., 2020). Additionally, the evolution of water column stratification plays a key role in promoting blooms with a clear signal in the North Atlantic which in fine may alter the regional carbon cycle. These dynamics, including the vertical variation of the summertime mixed-layer depth (Sallée et al., 2006), may reduce both light and nutrient availability, and also have implications for carbon export, a critical function of calcifying species. Knowing the impact of these blooms on the regional ocean carbon cycle, there is a clear interest in knowing the future evolution and implication as these weakening of the ocean carbon sink may compound with decline related to MHW events (Müller et al., 2025).

The changes observed in 2023 are an extreme signature of multi-annual variability superimposed on long-term trends. There is a need to disentangle the contributions of internal climate system variability, such as decadal variability, from the impacts of anthropogenic climate change. This will increase our capacity to assess extreme but plausible events such as the record SSTs in 2023-2024 (Terhaar et al., 2025) and anticipate their consequences. Advancing our understanding of these processes requires leveraging recent advances in attribution science (Stott et al., 2016; Ribes et al., 2020; Faranda et al., 2024), which have predominantly focused on terrestrial and atmospheric systems. Similar services for oceans, incorporating biogeochemical components, could be created. This effort could be developed by considering a combination of multi-scale observation networks capable of providing the initial conditions, and enhanced modelling frameworks that capture subsurface dynamics and multi stressor interactions to anticipate future changes and inform adaptive strategies for marine ecosystems (Gregg and Casey, 2007; Nissen et al., 2018; Krumhardt et al., 2019)."

---

## Referee Report (RR1)

**Review of "Exceptional 2023 marine heatwave reshapes North Atlantic coccolithophore blooms"**
**Guinaldo & Neukermans**

This review is based on egusphere-2025-1862-manuscript-version3

The authors present an observations-based study of coccolithophore blooms in two geographical domains, the Celtic Sea and the Baltic Sea, over the period 1998-2023, looking in particular for anomalies in 2023 corresponding to the marine heatwaves at that time.

Reviewers #1 and #2 were critical of many aspects in the original submission. They found many claims without references, and that several important terms and concepts were not well defined. Both reviewers described the paper as purely descriptive, contributing no new understanding of methods or concepts. In response, the authors have substantially revised their manuscript.

The revised paper appears to have addressed many of these criticisms. It is still a chiefly descriptive paper, but of a subject that is topical and relevant. As Reviewer #1 notes, it is a decent documentation of the event and as an analysis it does have quality. It is clear that a lot of work has gone into the study. The paper contributes to our knowledge of the impact of marine heat waves on ecosystems, a subject that is topical and that we know too little of. The impact on coccolithophores is important for carbon uptake, as discussed in the paper. The impact of MHW more generally on algal blooms is also important for higher trophic levels, the wider ecosystem, and for commercial interests such as aquaculture. The study gives useful confirmation that the environmental ranges defined by O'Brien et al for coccolithophore blooms do seem to apply even in these anomalous MHW conditions and in two separate regions.

There is a lot of material in the appendix. I would prefer to see some of this moved to the main paper. As a reader, it can be frustrating to have to skip to the appendix to view information that is important in understanding key points in the paper. In particular figures A4 and A6, if that is allowed.

Below are more detailed comments on specific parts of the paper.

Line 13: replace "forms" with "form"

Section 2.1:
This should mention the origin of the data sources used (SST, PAR, MLD). Details can be provided in the appendix but a brief mention is needed here.
For explanation of O'Brien *et al* we are told to look at A1.4. Lines 309-313 of A1.4 give a very short explanation which could be moved here so that the reader doesn't have to jump.

Lines 72, 73 and 90, 91: These have reminders of the ranges from O'Brien. Not needed.

Line 77: "later" would be better than "onward"

Lines 81-84: Strictly, the NAO is just a number and doesn't drive turbulent mixing. Strong positive values of the NAO are associated with storms that do drive mixing. References to the NAO in this section do need to make clear when they are talking about positive or negative values of this index.

Line 87: replace "relations" with "relationship"

Line 87,88: I suggest deleting "the consequences of"
Also, choose either "the persistent high-pressure system" or "persistent high-pressure systems" (no "the")

Line 97: "close to the normal winds" would be better without "the".

Line 104: "fall" is repeated. Unless EGU editorial policy recommends American English, this should be "autumn"?

Line 111 mentions correlations shown in the appendix, table A1. It's a very small table. Could these numbers please be moved into the main paper?

Line 118: "intrusion" implies that the coccolithophores have moved into the BS from the Atlantic. Is this what the authors mean? If so, it needs some evidence. If not, perhaps "expansion" would be a better word, as it's neutral over whether the coccolithophores moved there or bloomed *in situ*.

Line 120 and Figure 3 mention LOESS. As an acronym it needs expanding the first time it's used. Maybe even a citation such as Cleveland 1979?
https://doi.org/10.1080/01621459.1979.10481038

Line 126 talks about Atlantification. As the other reviewers noted, it would be useful to describe what this word means. I guess that water on both sides of the Polar Front is becoming more "Atlantic"? This might be worth mentioning, so that we don't view Atlantification as just being a northward shift in the Polar Front.

Caption for Figure 3: In the final sentence, replace "indicated" with "indicate".

Line 138: "hinder" isn't quite the right word. "disregard" would be better.

Line 143: replace "a shift" with "an eastward shift"

Line 144: I suggest replacing "front positions have" with "front position has"
Also, replace "another value close to the record high" with "a value close to the maximum"

Line 151: replace "revealed" with "reveals", "both" with "the two" (else you are saying that each region independently has dynamics that contrast with itself)

Line 156: Suggest delete "despite interannual variability"

Lines 156-157: replace "seems to imprint" with "gives"

Line 159: "durations" should be singular

Line 160: the 2nd "mid-June" needs deleting?

Lines 168-169: The sentence beginning "In both regions" doesn't read very well and needs rewriting. I suggest:
"Positive stratification anomalies were recorded in both regions in 2023, with the CS reaching record levels, which supported favourable conditions for *G. huxleyi*."

Line 174: replace "analyses" with "analysis"

Line 178: suggest deleting "oceanic"

Line 179: suggest deleting "dedicated"

Line 187: suggest deleting "within the 6-16°C range". Either that or add a range for PAR, so that SST and PAR are given consistently.

Lines 192-193: Suggest deleting the sentence "In addition…"
It feels out-of-place as this paper is not about wider ecosystem responses or adaptation plans and policies. If the authors did want to keep this sentence then the paper really needs more text on this subject.

Lines 200-202: Suggest deleting from "and combine" to the end of the sentence. Again, this opens a new topic that would need more text. For instance, can we really believe that numerical simulations of coccolithophore blooms will "accurately quantify the contributions of the respective processes"? I understand that it is a suggestion for further work, but I think it isn't needed and that the science isn't yet able to comply.

Line 205: suggest "due to masking by cloud cover" to make clear how cloud cover affects the satellite estimates

Lines 209-210: suggest replacing "vertically resolved" with "sub-surface" which has clearer meaning

Line 215: The sentence beginning "Additionally…" doesn't completely make sense after the word "Atlantic".

Lines 218-221: Do we know the impact of these blooms on the regional ocean carbon cycle? If not, you might delete from "Knowing" up to the first comma?

Line 222: suggest deleting "and reaching exceptional level". The next words ("are an extreme signature") give the same message and sound better.

Lines 245-246: suggest "…where blooms occur annually and marine heat waves resulted in…"

Line 270: "in a regular" should be "on a regular"

Line 278: Can the authors explain why March-April SSTs are used for estimating the Polar Front?

Lines 283:285: "To evaluate…"
No need to mention this evaluation unless you quote results from it.

Figure A9: I found this figure hard to interpret. Wouldn't a timeseries plot be clearer?

Table A1: Please explain that MJ and JAS indicate months.
All the values in the table have *** to indicate p value < 0.01
Suggest adding a comment that they are all significant at this level, and omitting the ***
Suggest also explaining why no values were calculated for BS for May-June.

---

## Referee Report (RR2)

The manuscript «Exceptional 2023 marine heat wave reshapes North Atlantic coccolithophore blooms» assesses the impact of North Atlantic coccolitophore blooms to both extreme events and long-term ocean warming. Using 25 years of satellite-derived particulate inorganic carbon data, the authors compare bloom dynamics in two regions; the Celtic Sea and the Barents Sea. As I highlighted in the first round, the study is relevant because it provides valuable insights into carbon cycle responses to both long-term and extreme climatic events.

I think that the authors have made a great effort to improve the manuscript from the previous iteration. My main complaints were that the article was primarily descriptive, and lacked clear indications of its contributions in terms of new data or methods. Additionally, some broad concepts were introduced without supporting information. I am glad to see that these flaws have been addressed. However, I still think that terms such as Atlantification and acidification are not quantified, and might be something to explicitly address or indicate. I point here especially to expanding the discussion of the impacts of ocean acidification, which can be beneficial to contextualize long-term declines in the productivity of these calcifying organisms.

I also believe that the paper would benefit from disentangle the contributions of transient extreme events from those of persistent global warming. This may not be within the scope of the study but I think it would be valuable to include something less descriptive but more quantitative (maybe attribution methods, whose importance are recognized by the authors to distinguish the internal climate variability vs anthropogenic climate change).

I don't have many specific comments: I believe the "Results" section should be "Results and Discussion". The paragraph from lines 170 to 175 seems like conclusions rather than results nor discussion, and should be removed or moved. Following lines 194 to 196, I also wonder if the title of the paper should be reworded to include the long-term effects. Line 31 suggests that internal variability and long-term warming trends are the drivers of mhw.

---

## Author Response (AR2)

**Reply to reviewers for manuscript entitled "Exceptional 2023 marine heatwave reshapes North Atlantic coccolithophore blooms"**

**Guinaldo & Neukermans**

**REV#2**

The manuscript «Exceptional 2023 marine heat wave reshapes North Atlantic coccolithophore blooms» assesses the impact of North Atlantic coccolithophore blooms to both extreme events and long-term ocean warming. Using 25 years of satellite-derived particulate inorganic carbon data, the authors compare bloom dynamics in two regions; the Celtic Sea and the Barents Sea.

As I highlighted in the first round, the study is relevant because it provides valuable insights into carbon cycle responses to both long-term and extreme climatic events. I think that the authors have made a great effort to improve the manuscript from the previous iteration. My main complaints were that the article was primarily descriptive, and lacked clear indications of its contributions in terms of new data or methods. Additionally, some broad concepts were introduced without supporting information. I am glad to see that these flaws have been addressed.

However, I still think that terms such as Atlantification and acidification are not quantified, and might be something to explicitly address or indicate. I point here especially to expanding the discussion of the impacts of ocean acidification, which can be beneficial to contextualize long-term declines in the productivity of these calcifying organisms.

I also believe that the paper would benefit from disentangle the contributions of transient extreme events from those of persistent global warming. This may not be within the scope of the study but I think it would be valuable to include something less descriptive but more quantitative (maybe attribution methods, whose importance are recognized by the authors to distinguish the internal climate variability vs anthropogenic climate change).

I don't have many specific comments: I believe the "Results" section should be "Results and Discussion". The paragraph from lines 170 to 175 seems like conclusions rather than results nor discussion, and should be removed or moved. Following lines 194 to 196, I also wonder if the title of the paper should be reworded to include the long-term effects. Line 31 suggests that internal variability and long-term warming trends are the drivers of mhw

We would like to thank the reviewer for their comments on our article, for the time they spent reviewing it in detail, and for their help in improving it.

We fully agree that disentangling the contribution of extreme events from the long-term anthropogenic trend would be highly valuable. However, such attribution analyses are

currently very challenging in a biogeochemical context, particularly for PIC or coccolithophore‑related variables. To our knowledge, formal detection–attribution frameworks have not yet been developed or applied robustly in ocean biogeochemistry, mainly because they require multi-decadal observational records combined with targeted model ensembles. For this reason, we remain limited to a descriptive interpretation of the respective roles of persistent warming and extreme events.

We propose a modified section in the discussion :

"Although attribution science has made substantial progress in recent years (Stott et al., 2016; Ribes et al., 2020; Faranda et al., 2024), these developments have focused primarily on terrestrial and atmospheric variables. In ocean biogeochemistry, formal attribution frameworks are still lacking, mainly because they require multi-decadal observations and dedicated model ensembles, which are not yet available for PIC variability. Developing attribution capabilities for marine biogeochemical systems would therefore require both multi-scale observation networks providing robust initial conditions and/or a solid observational "baseline" and improved modelling frameworks able to resolve subsurface dynamics and multi-stressor interactions (Gregg and Casey, 2007; Nissen et al., 2018; Krumhardt et al., 2019). Establishing such tools and datasets would be essential before the respective roles of internal variability, extreme events, and long-term anthropogenic forcing can be formally disentangled."

Regarding the definition of the concept of Atlantification we modified the section starting on Line.126:

"Two distinct processes contribute to this warming: long-term ocean temperature increase, especially pronounced at high latitudes, and the enhanced influence/inflow of Atlantic Water, commonly referred to as "Atlantification." (Årthun et al., 2012; Polyakov et al. 2017).  In the Barents Sea, Atlantification encompasses not only a northward shift of the Polar Front, but also the progressive warming, increase of salinity, loss of winter sea ice, and modification of stratification conditions of waters on both sides of the front. While these changes are strongest south of the front, modified Atlantic water increasingly reaches the northern, traditionally Arctic domain, particularly during ice-free winters (Årthun et al., 2012). To disentangle these contributions, we tracked the annual position of the Polar Front, a proxy for the influence of Atlantic water (Fig. A7a; Neukermans et al., 2018)."

Regarding the acidification, we propose to expand the discussion section as requested:

"Coccolithophores are microscopic, calcifying phytoplankton that contribute to marine primary production and global carbon cycling through both the organic carbon and carbonate pumps (Rost and Riebesell, 2004 ; Neukermans et al. 2023). Ocean acidification, driven by increased $CO_2$ uptake, reduces carbonate ion availability and lowers pH, creating challenging chemical conditions for calcifying organisms  (Riebesell et al 2000, Iglesias-Rodriguez et al 2008, Terhaar et al 2020}. Laboratory experiments show that impacts are dependent on the species. *Gephyrocapsa huxleyi* exhibits decreased calcification and lighter coccoliths under elevated $CO_2$, while other species may be more resilient but remain vulnerable to future acidification (Meyer et al 2015). Responses also

depend also on $CO_2$ enrichment which may allow partial compensation of calcification (Fukuda et al 2014). Global models project regionally heterogeneous effects, with some areas experiencing enhanced calcification due to carbon limitation alleviation, but a general decline is expected above ~600 µatm $CO_2$, with studies on long-term impacts suggesting progressive damages (Krumhardt et al. 2019, Tong et al. 2018}. Considering ocean acidification alongside warming and Atlantification provides essential context for interpreting the observed long-term declines in North Atlantic coccolithophore blooms."

References:

- Iglesias-Rodriguez, M. D., Halloran, P. R., Rickaby, R. E., Hall, I. R., Colmenero-Hidalgo, E., Gittins, J. R., ... & Boessenkool, K. P. (2008). Phytoplankton calcification in a high-CO2 world. *science*, *320*(5874), 336-340.
- Meyer, J., & Riebesell, U. (2015). Reviews and Syntheses: Responses of coccolithophores to ocean acidification: a meta-analysis. *Biogeosciences*, *12*(6), 1671-1682.
- Fukuda, S. Y., Suzuki, Y., & Shiraiwa, Y. (2014). Difference in physiological responses of growth, photosynthesis and calcification of the coccolithophore Emiliania huxleyi to acidification by acid and CO2 enrichment. *Photosynthesis research*, *121*(2), 299-309.
- Riebesell, U., Zondervan, I., Rost, B., Tortell, P. D., Zeebe, R. E., & Morel, F. M. (2000). Reduced calcification of marine plankton in response to increased atmospheric CO2. *Nature*, *407*(6802), 364-367.
- Tong, S., Gao, K., & Hutchins, D. A. (2018). Adaptive evolution in the coccolithophore Gephyrocapsa oceanica following 1,000 generations of selection under elevated CO 2. *Global Change Biology*, *24*(7), 3055-3064.

**New title proposition** : "Compounded effects of long-term warming and the exceptional 2023 marine heatwave on North Atlantic coccolithophore bloom dynamics"

We renamed the Results section "Results & Discussion". We also removed the lines 170-175.

**REV#3**

The authors present an observations-based study of coccolithophore blooms in two geographical domains, the Celtic Sea and the Baltic Sea, over the period 1998-2023, looking in particular for anomalies in 2023 corresponding to the marine heatwaves at that time.

Reviewers #1 and #2 were critical of many aspects in the original submission. They found many claims without references, and that several important terms and concepts were not well defined. Both reviewers described the paper as purely descriptive, contributing no new understanding of methods or concepts. In response, the authors have substantially revised their manuscript.

The revised paper appears to have addressed many of these criticisms. It is still a chiefly descriptive paper, but of a subject that is topical and relevant. As Reviewer #1 notes, it is a

decent documentation of the event and as an analysis it does have quality. It is clear that a lot of work has gone into the study. The paper contributes to our knowledge of the impact of marine heat waves on ecosystems, a subject that is topical and that we know too little of. The impact on coccolithophores is important for carbon uptake, as discussed in the paper. The impact of MHW more generally on algal blooms is also important for higher trophic levels, the wider ecosystem, and for commercial interests such as aquaculture. The study gives useful confirmation that the environmental ranges defined by O'Brien et al for coccolithophore blooms do seem to apply even in these anomalous MHW conditions and in two separate regions.

There is a lot of material in the appendix. I would prefer to see some of this moved to the main paper. As a reader, it can be frustrating to have to skip to the appendix to view information that is important in understanding key points in the paper. In particular figures A4 and A6, if that is allowed.

Thank you for this comment. We agree that Figures A4 and A6 contain important information, and we understand this comment. Because *Ocean Science Letters* has strict constraints on the length and structure of the main text, we need to confirm with the editor whether moving additional figures into the main manuscript is allowed.

In the following response, we accepted all suggestions unless a comment (in blue) indicated otherwise.

Below are more detailed comments on specific parts of the paper.

Line 13: replace "forms" with "form"

Section 2.1: This should mention the origin of the data sources used (SST, PAR, MLD). Details can be provided in the appendix but a brief mention is needed here. For explanation of O'Brien et al we are told to look at A1.4. Lines 309-313 of A1.4 give a very short explanation which could be moved here so that the reader doesn't have to jump.

To comply with the OS Letters guidelines, we prefer to keep the reference to the method in parentheses and keep the references as written in the manuscript.

Regarding the reference to Obrien's work, we noted the reviewer's comment. However, upon reading the two sections again, we feel that they complement each other and do not require any changes.

Section in the main text :

"To evaluate the impact of MHW on *G.huxleyi* blooms, we examine impacts on the three most influential environmental variables that characterize the ecological niches of coccolithophore species, namely SST, Photosynthetically Active Radiation (PAR), and the depth of the mixed layer (MLD), an indicator for the water column stratification (see Sect.A1.4; O'Brien et al 2016). For *G.huxleyi*, the optimal SST range was found to be situated between 6 and 16°C, optimal PAR between 35 and 42 Einstein.m−2.day−1}, and optimal MLD between 20 and 30 m (O'Brien et al., 2015). These ranges were extracted from the realized ecological niche of *G.huxleyi* (i.e. the environmental conditions under which it

can be observed) set up by (O'Brien et al., 2015), based on a global compilation of in situ measurements of coccolithophore species abundance and diversity (O'Brien et al., 2013)."

Section in Methods :

"Based on a global compilation of in situ measurements of coccolithophore species abundance and diversity (O'Brien et al., 2013), the realized ecological niche of G.huxleyi (i.e. the environmental conditions under which it can be observed) has been characterized (O'Brien, 2015). Out of seven environmental variables considered, O'Brien et al. (2016) showed that SST, PAR, and MLD were the most important variables influencing coccolithophore diversity. For G.huxleyi, the optimal SST range is situated between 6 and 16°C, optimal PAR between 35 and 42 Einstein.m−2.day−1, and optimal MLD between 20 and 30 m."

Lines 72, 73 and 90, 91: These have reminders of the ranges from O'Brien. Not needed.

In the first reviews received, reviewers suggested that references to conditions for blooms must be repeated for clarity. We therefore decided to maintain these references and reminders.

Line 77: "later" would be better than "onward"

Lines 81-84: Strictly, the NAO is just a number and doesn't drive turbulent mixing. Strong positive values of the NAO are associated with storms that do drive mixing. References to the NAO in this section do need to make clear when they are talking about positive or negative values of this index.

Thank you for this helpful comment. We agree that the NAO itself is an index and does not directly drive turbulent mixing. We have revised the text to clarify that it is the atmospheric conditions typically associated with positive (or negative) NAO phases that influence storminess, wind stress, and vertical mixing. The revised section now explicitly distinguishes between the NAO index and the physical mechanisms (e.g., westerlies, storm activity) that accompany its positive phase.

"In winter, the North Atlantic Oscillation (NAO) influences vertical turbulent mixing through the atmospheric conditions associated with its positive phase, which typically include enhanced westerlies and increased storm activity over the North Atlantic (Hurrell et al., 2003). Even at the northern edge of the North Atlantic, the BS atmospheric and oceanic internal variability responds to both positive and negative NAO conditions (Levitus et al., 2009; Chafik et al., 2015). In contrast, summer conditions favor the likelihood of high-pressure blocking systems over northern Europe (Rantanen et al., 2022; Rousi et al., 2022), characterized by weak winds and high solar radiation (Fig. A1, Fig. A2, Fig. A3)."

Line 87: replace "relations" with "relationship"

Line 87,88: I suggest deleting "the consequences of" Also, choose either "the persistent high-pressure system" or "persistent high-pressure systems" (no "the")

Line 97: "close to the normal winds" would be better without "the".

Line 104: "fall" is repeated. Unless EGU editorial policy recommends American English, this should be "autumn"?

Line 111 mentions correlations shown in the appendix, table A1. It's a very small table. Could these numbers please be moved into the main paper?

Thanks for this comment. We deleted the table and moved the numbers in the text.

"These levels were anomalously high in 2022 and 2023 (mean surface extent anomaly: 25 551 km2; maximum surface extent anomaly: 25 864 km2) and showed a significant correlation with spring–summer SSTs. Correlation coefficients reached 0.76 and 0.70 for mean and maximum surface extent, respectively, in relation to May–June SSTs, and 0.98 and 0.77 for mean and maximum surface extent, respectively, in relation to July–August–September SSTs (all p_value < 0.01)."

and

"In the BS, the increase in bloom extent was also strongly and significantly correlated with summer SSTs (Fig.A5b), with correlation coefficients of 0.95 and 0.96 for mean and maximum surface extent, respectively, in relation to July–August–September SSTs (p < 0.01). This highlights the role of warming in driving these changes."

Line 118: "intrusion" implies that the coccolithophores have moved into the BS from the Atlantic. Is this what the authors mean? If so, it needs some evidence. If not, perhaps "expansion" would be a better word, as it's neutral over whether the coccolithophores moved there or bloomed in situ.

We agree with this comment. We did not quantify the contribution of bio-advection (Oziel et al. 2020) and *in situ* proliferation. However, if we replace "intrusion" by "expansion" we will have a sentence containing two occurrences of the word "expansion." We therefore propose the following sentence:

"This reflects a northeastward expansion of coccolithophores in the BS linked to the shifting polar front."

Line 120 and Figure 3 mention LOESS. As an acronym it needs expanding the first time it's used. Maybe even a citation such as Cleveland 1979? https://doi.org/10.1080/01621459.1979.10481038

"Over the past 25 years, Locally Estimated Scatterplot Smoothing regression (LOESS, Cleveland, 1979) reveals significant positive trends in bloom extent"

Line 126 talks about Atlantification. As the other reviewers noted, it would be useful to describe what this word means. I guess that water on both sides of the Polar Front is becoming more "Atlantic"? This might be worth mentioning, so that we don't view Atlantification as just being a northward shift in the Polar Front.

Thank you for this comment. We agree that "Atlantification" should be defined more precisely. In the Barents Sea, it does not only correspond to a northward shift of the Polar Front, but also to an increasing inflow of Atlantic Waters in the sea. While the effects are strongest south of the front, Atlantic Water increasingly reaches the Arctic region, particularly during ice-free winters, affecting surface and subsurface conditions. We have clarified this in the text.

"Two distinct processes contribute to this warming: long-term ocean temperature increase, especially pronounced at high latitudes, and the enhanced influence/inflow of Atlantic Water, commonly referred to as "Atlantification. (Årthun et al., 2012; Polyakov et al., 2017). In the BS, Atlantification encompasses not only a northward shift of the Polar Front, but also the progressive warming, increase of salinity, loss of winter sea ice, and modification of stratification conditions of waters on both sides of the front. While these changes are strongest south of the front, modified Atlantic water increasingly reaches the northern, traditionally Arctic domain, particularly during ice-free winters (Årthun et al., 2012). To disentangle these contributions, we tracked the annual position of the Polar Front, a proxy for the influence of Atlantic water (Fig. A7a; Neukermans et al., 2018)."

Caption for Figure 3: In the final sentence, replace "indicated" with "indicate".

Line 138: "hinder" isn't quite the right word. "disregard" would be better.

Line 143: replace "a shift" with "an eastward shift"

Line 144: I suggest replacing "front positions have" with "front position has" Also, replace "another value close to the record high" with "a value close to the maximum"

Line 151: replace "revealed" with "reveals", "both" with "the two" (else you are saying that each region independently has dynamics that contrast with itself)

Line 156: Suggest delete "despite interannual variability"

Lines 156-157: replace "seems to imprint" with "gives"

Line 159: "durations" should be singular

Line 160: the 2nd "mid-June" needs deleting?

Lines 168-169: The sentence beginning "In both regions" doesn't read very well and needs rewriting. I suggest: "Positive stratification anomalies were recorded in both regions in 2023, with the CS reaching record levels, which supported favourable conditions for G. huxleyi."

Thanks. This sentence has been changed accordingly.

Line 174: replace "analyses" with "analysis"

Line 178: suggest deleting "oceanic"

Line 179: suggest deleting "dedicated"

Line 187: suggest deleting "within the 6-16oC range". Either that or add a range for PAR, so that SST and PAR are given consistently.

Lines 192-193: Suggest deleting the sentence "In addition..." It feels out-of-place as this paper is not about wider ecosystem responses or adaptation plans and policies. If the authors did want to keep this sentence then the paper really needs more text on this subject.

Lines 200-202: Suggest deleting from "and combine" to the end of the sentence. Again, this opens a new topic that would need more text. For instance, can we really believe that numerical simulations of coccolithophore blooms will "accurately quantify the contributions of the respective processes"? I understand that it is a suggestion for further work, but I think it isn't needed and that the science isn't yet able to comply.

Line 205: suggest "due to masking by cloud cover" to make clear how cloud cover affects the satellite estimates

Lines 209-210: suggest replacing "vertically resolved" with "sub-surface" which has clearer meaning

Line 215: The sentence beginning "Additionally..." doesn't completely make sense after the word "Atlantic".

Mention of impacts on regional carbon cycle in the next sentence so we remove the mention in this sentence. The new sentence is the following : "Additionally, the evolution of water column stratification plays a key role in promoting blooms with a clear signal in the North Atlantic."

Lines 218-221: Do we know the impact of these blooms on the regional ocean carbon cycle? If not, you might delete from "Knowing" up to the first comma?

Thank you for the comment. A few studies have attempted to  quantify  the impact of coccolithophore blooms on the regional carbon cycle, including their effects on surface $pCO_2$, air–sea $CO_2$ fluxes, primary production, calcification, and carbon export via the calcite ballast effect. This quantification remains challenging. To expand on  this point, we have revised the sentence and added key references documenting these impacts :

- Shutler, J. D., Land, P. E., Brown, C. W., Findlay, H. S., Donlon, C. J., Medland, M., ... & Blackford, J. C. (2013). Coccolithophore surface distributions in the North Atlantic and their modulation of the air-sea flux of CO 2 from 10 years of satellite Earth observation data. *Biogeosciences*, *10*(4), 2699-2709.
- Rigual Hernández, A. S., Trull, T. W., Nodder, S. D., Flores, J. A., Bostock, H., Abrantes, F., ... & Northcote, L. C. (2020). Coccolithophore biodiversity controls carbonate export in the Southern Ocean. *Biogeosciences*, *17*(1), 245-263.
- Delille, B., Harlay, J., Zondervan, I., Jacquet, S., Chou, L., Wollast, R., ... & Gattuso, J. P. (2005). Response of primary production and calcification to changes of pCO2 during experimental blooms of the coccolithophorid Emiliania huxleyi. *Global Biogeochemical Cycles*, *19*(2).

- Klaas, C. & Archer, D. E. Association of sinking organic matter with various types of mineral ballast in the deep sea: Implications for the rain ratio. *Glob. Biogeochem. Cycles* **16**, 63-1–63–14 (2002).

Section modified :

"Coccolithophore blooms can influence the regional ocean carbon cycling by modifying surface $pCO_2$ through the combined effect of primary production and calcification air–sea $CO_2$ exchange, and carbon export and deep particle fluxes through the calcite ballast effect (Shutler et al., 2013; Delille et al., 2005; Klaas & Archer, 2002; Rigual Hernández et al., 2020). Understanding and disentangling these influences on carbon cycling now and in the future is therefore crucial, especially as any potential long-term weakening of the ocean carbon sink may compound with short-term decline associated with MHW events (Muller et al 2025)."

Line 222: suggest deleting "and reaching exceptional level". The next words ("are an extreme signature") give the same message and sound better.

Lines 245-246: suggest "...where blooms occur annually and marine heat waves resulted in..."

Line 270: "in a regular" should be "on a regular"

Line 278: Can the authors explain why March-April SSTs are used for estimating the Polar Front?

As referenced in Neukermans et al (2018), the position of the SST front in March–April is a good indicator for the annual extent of Atlantic waters as the water column is vertically well mixed and the Atlantic waters are not yet stratified.

Lines 283:285: "To evaluate..." No need to mention this evaluation unless you quote results from it.

Thanks for this. Sentence modified : Vertical temperature and the stratification are derived from the Institute of Atmospheric Physics (IAP) observation-based temperature/salinity fields at 1°x1° horizontal resolution and 41 vertical levels from 1-2000m and a monthly resolution from January 1940 to September 2023 were used.

Figure A9: I found this figure hard to interpret. Wouldn't a timeseries plot be clearer?

Thank you for this suggestion. In this figure, we show the annual maxima of PIC, so the temporal resolution is already yearly rather than continuous. For this type of discrete annual information, we believe a barcode representation is more appropriate than a lineplot.

Table A1: Please explain that MJ and JAS indicate months. All the values in the table have *** to indicate p value < 0.01 Suggest adding a comment that they are all significant at this level, and omitting the *** Suggest also explaining why no values were calculated for BS for May-June.

In a previous comments, Rev#3 asked to remove this table and add the numbers in the text.